# Anisotropic thermal conductivity of antigorite along slab subduction impacts seismicity of intermediate-depth earthquakes

Yu-Hsiang Chien[1,2,3], Enrico Marzotto [4,5] ✉, Yi-Chi Tsao[2] & Wen-Pin Hsieh [2,6] ✉

Double seismic zones (DSZs) are a feature of some subducting slabs, where intermediate-depth earthquakes (~70–300 km) align along two separate planes. The upper seismic plane is generally attributed to dehydration embrittlement, whereas mechanisms forming the lower seismic plane are still debated. Thermal conductivity of slab minerals is expected to control the temperature evolution of subducting slabs, and therefore their seismicity. However, effects of the potential anisotropic thermal conductivity of layered serpentine minerals with crystal preferred orientation on slab's thermal evolution remain poorly understood. Here we measure the lattice thermal conductivity of antigorite, a hydrous serpentine mineral, along its crystallographic b- and c-axis at relevant high pressure-temperature conditions of subduction. We find that antigorite's thermal conductivity along the c-axis is ~3–4 folds smaller than the b-axis. Our numerical models further reveal that when the low-thermal-conductivity c-axis is aligned normal to the slab dip, antigorite's strongly anisotropic thermal conductivity enables heating at the top portion of the slab, facilitating dehydration embrittlement that causes the seismicity in the upper plane of DSZs. Potentially, the antigorite's thermal insulating effect also hinders the dissipation of frictional heat inside shear zones, promoting thermal runaway along serpentinized faults that could trigger intermediate-depth earthquakes.

Slab subduction is the main driving force of plate tectonics, which introduces complex chemical heterogeneities in Earth's interior[1]. During subduction, the rocks within the slab are subjected to increasing pressure (P) and temperature (T) conditions, which alter their physical and chemical properties with impacts on the thermo-chemical evolution of the slab, and the geodynamics of the subduction zone and surrounding mantle. In particular, the temperature profile of the subduction zone is key to influence the stability of the minerals and rocks within it, as well as its geological processes, e.g., arc volcanism and magmatism[2] and earthquakes[3]. Therefore, quantifying the tem-

[1]Earth System Science Program, Taiwan International Graduate Program (TIGP), Academia Sinica and National Central University, Taipei, Taiwan, ROC. [2]Institute of Earth Sciences, Academia Sinica, Taipei, Taiwan, ROC. [3]College of Earth Sciences, National Central University, Taoyuan, Taiwan, ROC. [4]Helmholtz Center Potsdam, GeoForschungsZentrum (GFZ), Potsdam, Germany. [5]Institute of Geosciences, University of Potsdam, Karl-Liebknecht-Straße 24-25, 14476 Potsdam, Germany. [6]Department of Geosciences, National Taiwan University, Taipei, Taiwan, ROC. ✉e-mail: marzotto@gfz-potsdam.de; wphsieh@earth.sinica.edu.tw

**Fig. 1 | High pressure thermal conductivity of single-crystal antigorite along [010] (b-axis, black circles) and [001] (c-axis, red squares) to 13 GPa at room temperature.** Vertical bar at each datum point represents the data uncertainty. The thermal conductivity anisotropy ($\Lambda^{010}/\Lambda^{001}$) is ~2.5–4.3 between ambient and 7 GPa. Note that the difference between $\Lambda^{010}$ and $\Lambda^{001}$ reaches a maximum of 7.3 W m$^{-1}$ K$^{-1}$ around 7 GPa, where a displacive phase transition occurs. Literature results for polycrystalline antigorite at ambient pressure[31] (open blue circle) and to 8.5 GPa[32] (open green stars) are plotted for comparison.

perature evolution of a slab along subduction is fundamental to understand a large variety of geophysical, geodynamical, and geochemical phenomena around a subduction zone.

Interestingly, in some subduction zones, earthquakes occur at intermediate-depth (~70–300 km) along two distinct planes separated by ~20–40 km, known as double seismic zones (DSZs), see refs. 4–6 for example features of DSZs, where the separation of two seismic planes correlates with the age of a slab, i.e., with its temperature and thickness at the trench. Since its first observation in northeast Japan[7], several mechanisms have been proposed to explain the origin of DSZs, including dehydration embrittlement[6,8–13], fluid-related embrittlement[14], transformation faulting[15], plastic shear instability[16,17], grain size reduction[18], and thermal runaway[12,19–22]. Of particular importance are the dehydration embrittlement and thermal runaway, as they are considered the two major mechanisms for intermediate-depth seismicity[12,19]. Dehydration embrittlement is triggered by the presence of free water $H_2O$ in the interstitial pores of the rock, which reduces the differential stress required for brittle fracture. Most of the water released at depth originates from the breakdown of antigorite[23,24], a high-pressure polymorph of serpentine minerals that contains large amounts of water (~12 wt%) and serves as the major hydrous mineral in subduction zones. Antigorite, however, does not survive at the high T condition in Earth's interior, and breaks down at ~700–1000 K[23,24]. This temperature is typically reached inside the slab at about 70–300 km depth, depending on the P-T path of the slab[24,25]. As a result, the intermediate-depth seismicity at the upper plane of a DSZ has often been attributed to the temperature-induced breakdown and dehydration of antigorite, see, e.g., refs. 3,6,26. The mechanism that causes seismicity at the lower plane, however, remains debated.

Thermal conductivity of minerals within a subduction zone plays key roles in controlling the heat flow and the temperature profile in the region. Thus, precise determination of such critical property at relevant P-T conditions would offer crucial insights into the thermal structure and dynamics of the subduction zone, as well as the fate of antigorite, with potential implications for the formation and distribution of intermediate-depth earthquakes. Lattice thermal conductivity or diffusivity of mantle minerals under extreme P-T conditions have been studied for decades. Precise measurements, however, are challenging due to the difficulty and limitation of previous conventional experimental techniques. As a result, very few experimental data under relevant mantle conditions were available and the measurement

accuracy was insufficient. Recent successful combinations of time-resolved optical techniques with diamond anvil cells (DACs) have enabled precise measurements of minerals' thermal conductivity under high P-T conditions, offering novel insights to the complex thermal evolution and dynamics in Earth's deep interior[27–30].

Prior studies on the thermal conductivity of antigorite have been limited to polycrystalline samples at relatively low pressure regime[31,32]. Given its layered crystal structure, antigorite is characterized by a strong anisotropy in its elastic constants along different crystal orientations[33–36]. Since the lattice heat transport scales with the elastic constants[37], antigorite's thermal conductivity is expected to be anisotropic as well. Such hypothesis, however, has not been experimentally tested. On the other hand, serpentinites are featured by a weak rheology and, upon shear deformation during slab subduction, serpentine crystals tend to develop a crystal preferred orientation (CPO), in which the crystal layers are parallel to the shear plane, i.e., antigorite's [001] direction orients perpendicularly to slab's dip direction[38,39]. As discussed above, the breakdown depth of antigorite during subduction depends on slab's temperature evolution, which could be critically controlled by antigorite's thermal conductivity. Moreover, previous thermomechanical modelings, e.g., refs. 6,9, have revealed the important roles that the slab age and convergence velocity play on antigorite's dehydration profile within a subducting slab, while the antigorite's thermal conductivity has often been assumed as a constant (e.g., ~3–4 W m$^{-1}$ K$^{-1}$), regardless of its crystal orientation and P-T conditions. The potential thermal conductivity anisotropy and CPO of antigorite during subduction would result in distinct thermal conductivity along a direction perpendicular and parallel to the slab's dip direction. Understanding the effects of P, T, and CPO on antigorite's thermal conductivity along subduction is thus critically needed, as it would significantly advance numerical modeling of the temperature profile in subduction zones.

In this work, we experimentally show a strong thermal conductivity anisotropy in antigorite under relevant P-T conditions along subduction. Our numerical simulations on the two-dimensional (2D) heat diffusion and temperature profile within a subducting slab indicate that antigorite's anisotropic thermal conductivity with CPO promotes dehydration embrittlement and potentially thermal runway, which could play key roles in triggering intermediate-depth earthquakes.

## Results

### Lattice thermal conductivity of antigorite at high pressure and room temperature

We used ultrafast time-domain thermoreflectance (TDTR) coupled with DAC to precisely measure the lattice thermal conductivity of single-crystalline natural antigorite to ~13 GPa at room temperature. The TDTR is an ultrafast pump-probe spectroscopy for high-precision thermal conductivity measurements at high pressures and wide range of temperatures, see, e.g., refs. 27,30,40 and Methods for details. Figure 1 shows the effect of pressure on the thermal conductivity of single-crystal antigorite along the [010] (in-plane b-axis, $\Lambda^{010}$, black circles) and [001] (cross-plane c-axis, $\Lambda^{001}$, red squares) orientation at room temperature. At ambient conditions, the $\Lambda^{010}$ is ~4.6 W m$^{-1}$ K$^{-1}$, ~4 times larger than the $\Lambda^{001}$ (~1.1 W m$^{-1}$ K$^{-1}$). Upon compression, $\Lambda^{010}$ and $\Lambda^{001}$ both increase with pressure and reach 10.0 (±1.9) and 2.7 (±0.4) W m$^{-1}$ K$^{-1}$, respectively, around 7 GPa, where the difference between them achieves a maximum value of 7.3 W m$^{-1}$ K$^{-1}$. In other words, the thermal conductivity anisotropy ($\Lambda^{010}/\Lambda^{001}$) at ambient conditions is ~4.3 and becomes less anisotropic as the anisotropy progressively decreases to ~2.5 at P ~ 4–5 GPa; afterwards the anisotropy slightly increases to ~3.7 at 7 GPa. Since thermal conductivity is an ensemble contribution of heat transport that involves heat capacity, elastic constant, and phonon mean-free-path of all the available phonon modes[37], such trend with pressure can be primarily accounted for

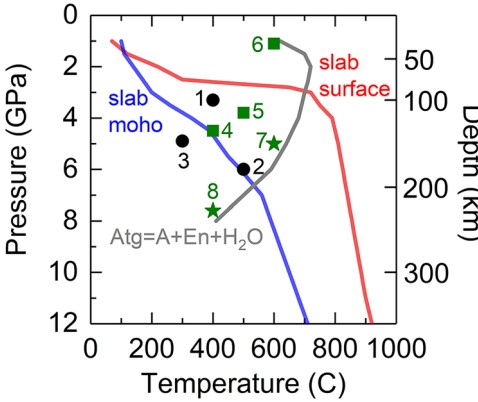

**Fig. 2 | P-T conditions for each experimental run (labeled by a number) on the measurement of antigorite's thermal conductivity.** P-T profiles for a typical subducting slab surface (red curve) and slab Moho (blue curve) as well as for the phase boundary (gray curve) of antigorite (Atg) decomposition into phase A (A), enstatite (En), and water are taken from ref. 3 and plotted for reference.

**Table 1 | High pressure-temperature conditions and the thermal conductivity data for antigorite**

| Measurement Run | $P$ (GPa) | $T$ (C) | Crystal axis | $\Lambda$ (W m$^{-1}$ K$^{-1}$) |
|---|---|---|---|---|
| 1 | 3.3 | 400 | b | 3.3 |
| 2 | 6 | 500 | b | 4.3 |
| 3 | 4.9 | 300 | b | 4.2 |
| 4 | 4.5 | 400 | c | 1.6 |
| 5 | 3.8 | 500 | c | 1.4 |
| 6 | 1.1 | 600 | c | 1 |
| 7 | 5 | 600 | c | 1.4 |
| 8 | 7.6 | 400 | c | 2 |

by the pressure evolution of the elastic constant along the in-plane and cross-plane direction, see, e.g., ref. 36. Note that the $\Lambda^{010}$ at 7 GPa is larger than the thermal conductivity of olivine (~5.1–8.3 W m$^{-1}$ K$^{-1}$)[41,42], the major mineral phase in the upper mantle. Interestingly, at P > 7 GPa the difference between $\Lambda^{010}$ and $\Lambda^{001}$ reduces progressively: the $\Lambda^{010}$ decreases considerably with pressure, while the $\Lambda^{001}$ reaches a plateau of ~3.0–3.5 W m$^{-1}$ K$^{-1}$ between 8 and 13 GPa, making them approach with each other. The occurrence of the dramatic change in the trend of $\Lambda^{010}$ with pressure around 7 GPa coincides with a displacive phase transition observed in antigorite[24,43,44], where pressure induces the distortion of Si-O tetrahedral and octahedral sheets[45].

In Fig. 1 we also plotted literature results for polycrystalline antigorite at ambient pressure[31] (open blue circle) and to 8.5 GPa[32] (open green stars), respectively, which are bracketed by our b- and c-axis data at P < 4 GPa. Nevertheless, the data by ref. 32. (open green stars) appear to be nearly a constant, rather than a typically increasing trend upon compression, and become comparable to our c-axis data at P ~ 4–9 GPa. We are not aware of the detailed experimental conditions in Osako's study, and thus could only propose that their polycrystalline sample may have been progressively oriented preferentially to near the c-axis under their compression conditions. By contrast, to check the potential variation of crystal orientation during our compression conditions, we have re-characterized the orientation from the quenched samples out of the DAC using electron backscattered diffraction or X-ray diffraction (in situ measurements of crystal orientation and thermal conductivity at high pressures are currently not available). We found that the orientation remained essentially the same, suggesting that the minor variation, if there is, of crystal orientation is expected to have minor effects on the uncertainty of our antigorite's thermal conductivity.

Note that we have also measured the thermal conductivity along the a-axis, $\Lambda^{100}$, at ambient conditions, which is ~4.5 W m$^{-1}$ K$^{-1}$ and close to that of the b-axis. Since thermal conductivity scales with the square of sound velocity[37], such behavior can be understood by the similar elastic constant and sound velocity along the b- and a-axis, even at high pressures[34,46]. Therefore, even though we did not measure the pressure dependence of $\Lambda^{100}$, we expect it to be similar to $\Lambda^{010}$ and much larger than the $\Lambda^{001}$. As we show later in the sections of numerical modeling and discussions, what critically impacts slab's thermal evolution is the thermal insulating effect by the low value of $\Lambda^{001}$, and the strong thermal conductivity anisotropy. Lack of data for $\Lambda^{100}(P, T)$ does not influence our conclusions of this study.

## Thermal conductivity at simultaneous high pressure and temperature conditions

To quantify the combined effects of pressure and temperature on the thermal conductivity of antigorite, we further performed simultaneous high P-T measurements. Figure 2 and Table 1 summarize the P-T conditions for each measurement run and its thermal conductivity along a specific crystal orientation, which allow us to track how the antigorite's thermal conductivity changes along slab subduction. Typical P-T profiles at slab surface and Moho are plotted as red and blue curve, respectively, in Fig. 2 from ref. 3. To quantify the temperature dependence of thermal conductivity, we compared our data at a given pressure and room temperature (Fig. 1) with those at a similar pressure and high temperatures (Fig. 2). For instance, we considered $\Lambda^{001}$ at 5 GPa and room temperature (Fig. 1), as well as measurement run 4 and 7 (Fig. 2). For simplicity, if we assume the $\Lambda$ of antigorite can be phenomenologically modeled as $\Lambda(T) = \Lambda_0 T^n$, where $\Lambda_0$ is a normalization constant at room temperature, then the temperature exponent n is determined by fitting a linear regression slope in the ln$\Lambda$-lnT plot. We found that n ~ −0.55 (±0.01) along the c-axis and n ~ −0.58(±0.06) along the b-axis. Considering our data uncertainty of ~15%, such dependences are reasonably in line with the typical temperature dependence of $T^{-0.5}$ for minerals[27,40,47,48]. The combined P-T dependences indicate that during slab subduction, antigorite's thermal conductivity anisotropy ($\Lambda^{010}/\Lambda^{001}$) remains at ~3–4 until ~230 km depth. Such critical finding is employed to model the thermal evolution of a sinking slab containing antigorite with CPO, see the numerical modeling section below.

## Numerically modeling the thermal evolution of a sinking slab

Our experiments reveal that antigorite's thermal conductivity is strongly anisotropic even at high P-T conditions ($\Lambda^{010} \sim 3\Lambda^{001}$). Potentially, such thermal conductivity anisotropy could induce temperature anomalies within a sinking slab, thus influencing the dehydration depth of antigorite, as well as promoting the onset of thermal runway. Therefore, investigating the large-scale effects of antigorite's thermal conductivity anisotropy would provide critical insights to the mechanisms responsible for the intermediate-depth earthquakes. For this purpose, we further performed 2D numerical modeling to compute the temperature evolution of a sinking slab containing antigorite (see Methods section). The full description of the model is reported in Supplementary Information: physical and numerical model (Note S1–S2, Fig. S4–S8, Table S1), thermodynamic parameters (Note S3–S5, Fig. S9, Tables S2–S3), model limitations (Note S6), and code benchmark (Note S7, Figs. S10–S11, Table S7). In total, we run 10 different models divided in three sets (Table S4). In the first set, used as a reference, we assumed that the hydrous layer was made entirely of dry olivine (model 1) or wet olivine (model 2). In the other two sets, we considered the [001] direction of antigorite (i.e., the c-axis) as the "insulating direction". We tested two ideal end-member

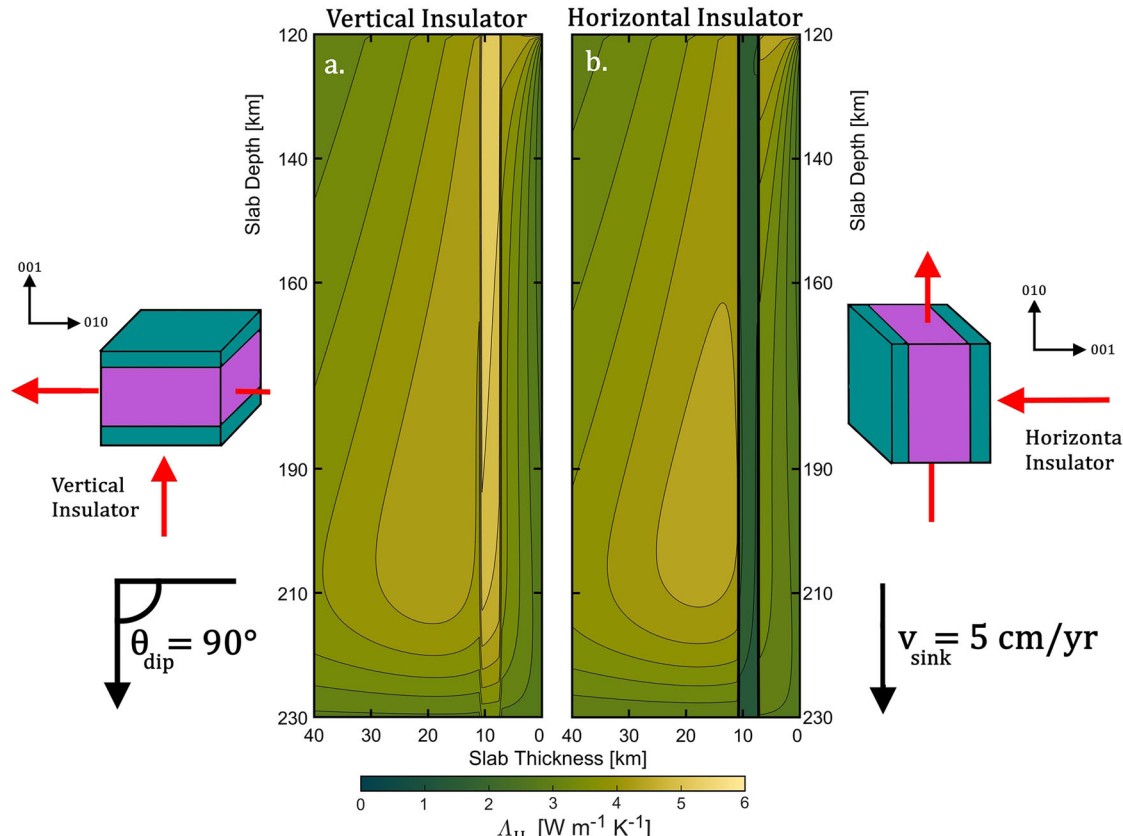

**Fig. 3 | Example 2D horizontal thermal conductivity fields $\Lambda_H$.** (**a**) shows model 6 ($\varphi_{Atg}$ = 1.0, vertical insulator) and (**b**) shows model 10 ($\varphi_{Atg}$ = 1.0, horizontal insulator). For illustration purpose, we only show the top 40 km of the slab, where slab surface starts at 0 km. In each plot we used the Scientific Color Maps[I]: low $\Lambda$ with shades of green, while high $\Lambda$ with shades of yellow. Note the 3-km-thick discontinuity of thermal conductivity corresponds to the serpentinized hydrous layer (7-10 km inside the slab). The two boxes on the sides represent the two orientation configurations of antigorite used in the model: the teal color indicates slow heat propagation, whereas the purple color indicates fast heat propagation. The heat flows are represented as red arrows. In (**a**), antigorite serves as a vertical insulator with the [001] in vertical direction. In (**b**), antigorite serves as a horizontal insulator with the [001] in horizontal direction. In the models, slabs subducted vertically with a constant sinking velocity of $v_{sink}$ = 5 cm yr$^{-1}$ and a dip angle $\theta_{dip}$ = 90 $_{\circ}$ see Methods section.

configurations: vertical insulator with antigorite's [001] direction parallel to the slab dip (models 3–6, Fig. 3a), and horizontal insulator with antigorite's [001] direction perpendicular to the slab dip (models 7–10, Fig. 3b), see Methods section for details. The assumption of perfect alignment of single-crystalline antigorite along the slab dip angle and the potential effects of grain boundary are fully discussed in the Supplementary Information Note S8, Fig. S13, and Table S8, and Note S9, respectively. In set 2 and 3, antigorite fraction $\varphi_{Atg}$ was increased to simulate different degrees of serpentinization in the slab: 0.1 (models 3,7); 0.3 (models 4,8); 0.5 (models 5, 9); 1.0 (models 6,10). Figure 4 illustrates two examples of 2D temperature fields from model 1 (Fig. 4a) and model 10 (Fig. 4b) and their temperature difference (Fig. 4c).

We monitored the temperature T (Fig. 4) and the heat flux Q (Fig. S12) in three specific planes from the slab surface (0 km): 7 km (external side of the hydrous layer), 9 km (inside the hydrous layer) and 11 km (internal side of the hydrous layer). In the horizontal insulator configuration (Fig. 3b), the thermal energy coming from the hot ambient mantle is not efficiently transported toward the cold core of the slab, accumulating at the interface between the oceanic crust and the insulating layer (e.g., Fig. 4c). Consequently, the temperature at the base of the crust (7 km) in the horizontal insulator configuration ($T_{7km}^{001}$, Fig. 5c teal dashed lines) is higher than the temperature in the vertical insulator configuration ($T_{7km}^{010}$, Fig. 5c purple dotted lines). Moreover, in the horizontal insulator configuration (Fig. 3b), the heat flow toward the cold core of the slab is hampered for the whole thickness of the hydrous

layer, and thus its internal temperature is higher than the vertical insulator configuration: $T_{9km}^{010} < T_{9km}^{001}$ (Fig. 5b). The insulation effect caused by the hydrous layer with antigorite in the horizontal insulator configuration is shown in Fig. 5a, where the temperature at 11 km from the slab surface is lower than the vertical insulator configuration: $T_{11km}^{010} > T_{11km}^{001}$. The temperature differences between the two configurations (models 7–10 vs. models 3–6) range between $+20\ K < (T_{7km}^{7-10} - T_{7km}^{3-6}) < +230\ K$ (Fig. 5c), $+10\ K < (T_{9km}^{7-10} - T_{9km}^{3-6}) < +100 K$ (Fig. 5b), and $-3K < (T_{11km}^{7-10} - T_{11km}^{3-6}) < -60\ K$ (Fig. 5a).

We compared these results also with the two reference models: 1 (dry olivine, $\Lambda^{DryOl}$) and 2 (wet olivine, $\Lambda^{WetOl}$). The thermal conductivity of wet olivine containing ~7000 wt. ppm water is 30–40% lower than its dry counterpart[41], and thus the hydrous layer in model 2 acts as a thermal insulator compared to model 1 (solid green lines in Fig. 5): $(T_{7km}^{WetOl} - T_{7km}^{DryOl}) < 80\ K$ (Fig. 5c) and $(T_{11km}^{WetOl} - T_{11km}^{DryOl}) > -30\ K$ (Fig. 5a). The vertical insulator configuration (models 3–6) does not produce a significant temperature difference compared to the dry reference even in the extreme case of complete serpentinization (Fig. 5): $-6\ K < T_{7km}^{3-6} - T_{7km}^{DryOl} < -60\ K$; and $+1.5\ K < T_{11km}^{3-6} - T_{11km}^{DryOl} < +11\ K$. In the horizontal insulator configuration, however, the temperature difference increases with the degree of serpentinization, and reaches its maximum in the extreme case of complete serpentinization: $+15\ K < (T_{7km}^{7-10} - T_{7km}^{DryOl}) < +170\ K$ at the external side (Fig. 5c) and $-3\ K < (T_{11km}^{7-10} - T_{11km}^{DryOl}) < -65\ K$ at the internal side

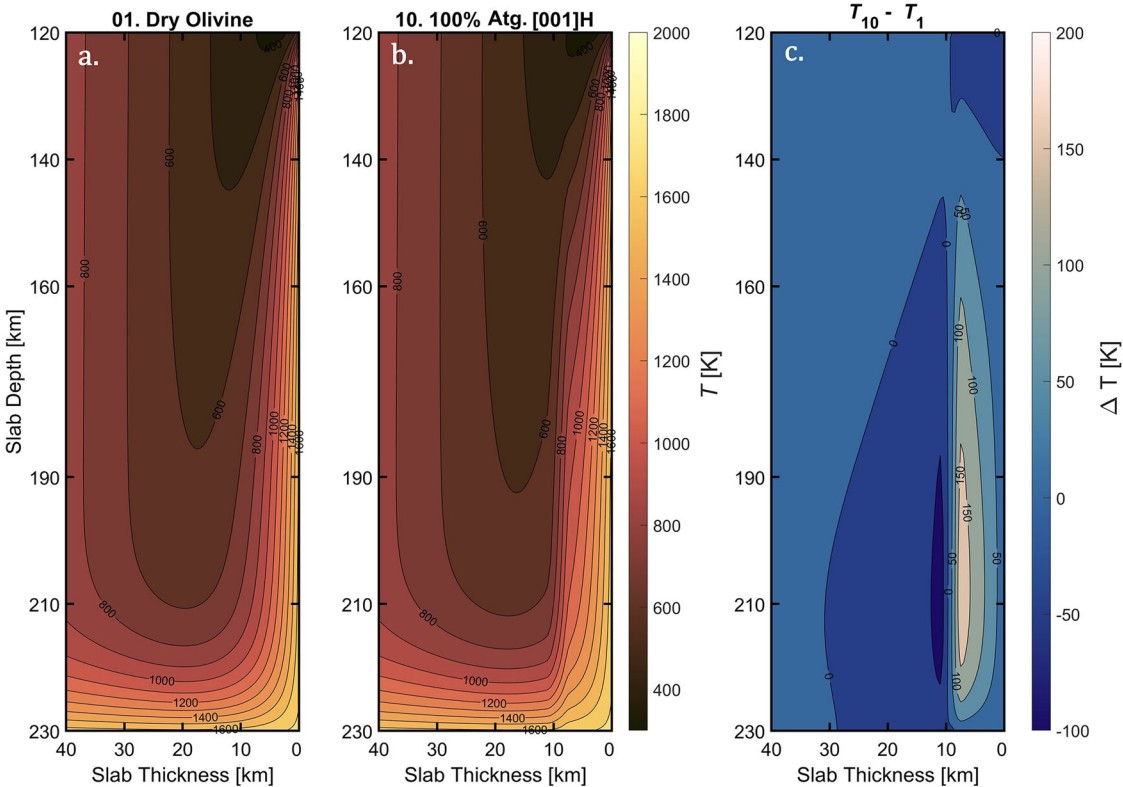

**Fig. 4 | Examples of 2D temperature field. (a)** shows model 1 ($\varphi_{Atg} = 0$; $\Lambda_H^{DryOl}$) and **(b)** shows model 10 ($\varphi_{Atg} = 1$; $\Lambda_H^{001}$). For illustration purpose, we only show the top 40 km of the slab, where slab surface starts at 0 km. In each plot we used the Scientific Color Maps[1] with low temperature in dim colors, while high temperature in bright colors. **(c)** Temperature difference $\triangle T$ ($K$) between model 10 and model 1.

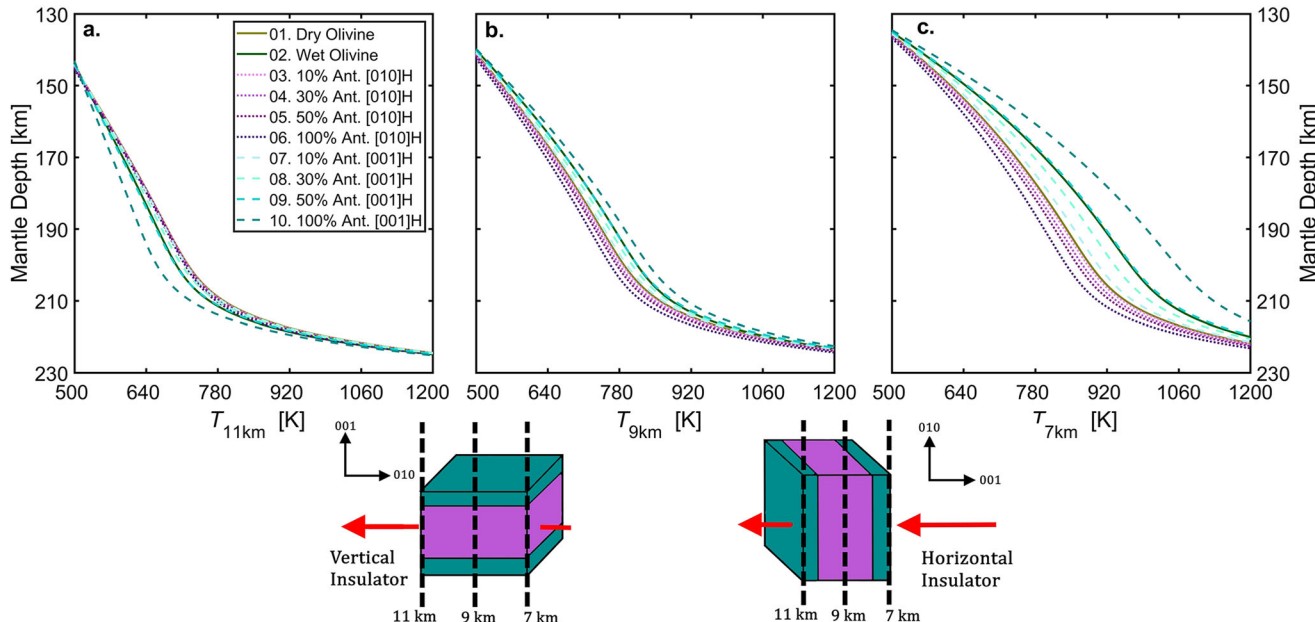

**Fig. 5 | Final temperature profiles of each slab at the end of the simulation.** Each subplot illustrates the temperature profile extracted at a different depth from the slab surface (dashed black lines). **a** $T_{11km}$ in the internal side of the hydrous layer (11 km from slab surface). **b** $T_{9km}$ inside the hydrous layer (9 km from slab surface). **c** $T_{7km}$ in the external side of the hydrous layer (7 km from slab surface). Each line indicates the $T$ profile of one model: green solid lines for set 1 as references, dotted purple lines for set 2 with vertical insulator, and dashed teal lines for set 3 with horizontal insulator.

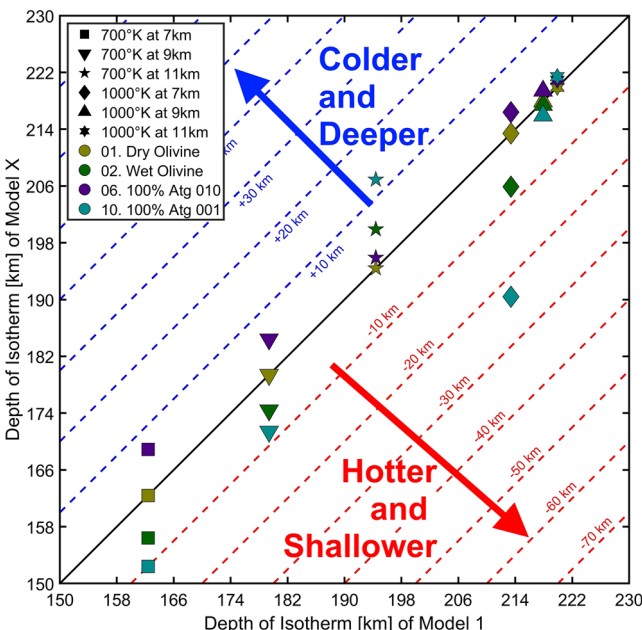

**Fig. 6 | Comparison of the maximum depth of 700 K and 1000 K isotherms between the reference model 1 ($\varphi_{Atg} = 0$; $\Lambda_H^{DryOl}$), in the x-axis, and all other models in the y-axis.** For simplicity we plotted only the models of 1, 2, 6, and 10 (all data are reported in Table S5). The marker color represents a given model with olive green for model 1, green for model 2, purple for vertical insulator, and teal for horizontal insulator. The marker shape represents the maximum depth of a given isotherm at a given depth inside the slab: squares, down-pointing triangles, and five-pointed stars for 700 K isotherm at 7 km, 9 km, and 11 km inside the slab, respectively; diamonds, up-pointing triangles, and six-pointed stars for 1000 K isotherm at 7 km, 9 km, and 11 km inside the slab, respectively. The solid black line represents the 1 : 1 correlation between the two models, where $D_{iso}^{DryOl} = D_{iso}^X$. We computed the difference between the models as $\triangle D_{iso} = D_{iso}^X - D_{iso}^{DryOl}$. The dashed blue lines on the upper left side indicate deeper isotherms due to colder conditions ($\triangle D_{iso} > 0$). The dashed red lines on the lower right side indicate shallower isotherms due to hotter conditions ($\triangle D_{iso} < 0$). Note that in the model 10 (horizontal insulator - teal) the maximum depths of the 700 K and 1000 K isotherms are shallower than the reference model 1 on the external side of the hydrous layer (7 km), whereas they are deeper on the internal side (11 km).

(Fig. 5a). Figure 5 indicates that a serpentinization degree of 50% (model 9) causes a temperature difference similar to the wet olivine case (model 2), where the oceanic crust is ∼ 80 K hotter than the dry reference (model 1). Given the limited water storage capacity of olivine in the upper mantle (only about few hundreds wt. ppm[49]) with relatively high thermal conductivity, antigorite with [001] direction would be a key mineral phase to act as a thermal blanket in the shallow upper mantle.

We further evaluated the effect of the thermal insulating layer on antigorite's breakdown depth (Fig. 6), which depends on the slab's internal temperature[25]. Along subduction the breakdown temperature of antigorite lays between $700 < T < 1000$ K[23,24], and therefore we computed the maximum depth of the 700 K and 1000 K isotherms ($D^{700}$ and $D^{1000}$) inside the slab (Table S5). When the antigorite acts as a horizontal insulator (teal colors in Fig. 6), the temperature of the external side of the hydrous layer ($T_{7km}$ – teal squares and diamonds in Fig. 6) is higher, and hence the two isotherms shift to shallower depths compared to the dry reference (model 1): $-1 < \triangle D_{7km}^{700} \leq -10$ km and $-1 < \triangle D_{7km}^{1000} < -23$ km. Inside the hydrous layer the trend is the same ($T_{9km}$ - down- and up-pointing triangles in Fig. 6): $-1 < \triangle D_{9km}^{700} < -10$ km and $0 < \triangle D_{9km}^{1000} < -3$ km. On the other hand, the temperature on the internal side of the hydrous layer ($T_{11km}$, five-pointed and six-pointed stars in Fig. 6) is lower, and the two isotherms survive at greater

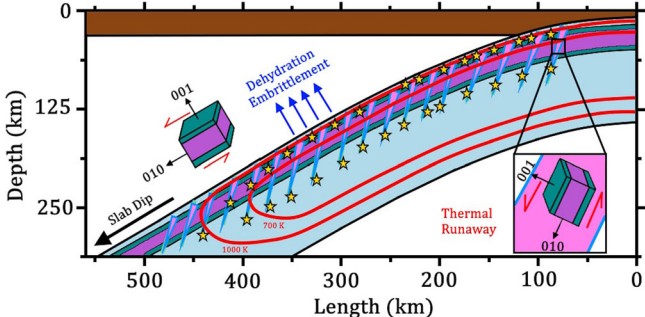

**Fig. 7 | Schematic drawing summarizing the effects of antigorite's thermal conductivity anisotropy combined with its CPO.** In the slab, antigorite is present in a layer of serpentinized rocks produced by hydrothermal circulation (teal and purple layers) and along the normal faults formed at the trench-rise system (light blue and pink sharp triangles). Antigorite's strong CPO aligns with the [001] direction normal to the shear stress. Once the slab temperature reaches ~700–1000 K (two red isotherm curves), dehydration embrittlement due to the breakdown of antigorite triggers earthquakes (yellow stars) that can form the upper seismic plane of a DSZ. Potentially, antigorite can trigger thermal runaway within the pre-existing serpentinized faults by promoting the accumulation of frictional heat within shear fault zones due to its thermal blanketing effect. These two mechanisms are likely the most plausible mechanisms responsible for intermediate-depth seismicity inside a slab.

depths: $+1 < \triangle D_{11km}^{700} < +13$ km and $0 < \triangle D_{11km}^{1000} < +2$ km. Consequently, the breakdown depth of antigorite at the external side of the insulating layer is ~15 km shallower, while that at the internal side of the insulating layer is only mildly affected.

## Discussion

The hydrothermal circulation active at the slow-spreading ridges can reach the bottom of the oceanic crust, and thus produce a few-km thick layer of serpentinized rocks within the lithospheric mantle[50,51]. Moreover, additional serpentinization occurs at the trench-rise system, where the bending of the oceanic lithosphere creates a network of normal faults along which seawater percolates down to ~20 km from the slab surface[52–55]. Potentially, the unbending of the slab in the mantle could generate a pressure gradient sufficient to pump seawater toward the inner portion of the slab, and to push the serpentinization front down to ~40 km from the slab surface[56]. The occurrence of this serpentinization process, however, is still under debate. In certain regions (e.g., Lesser Antilles), the seawater alteration of the oceanic lithosphere can produce a 3-km-thick layer of hydrous rocks containing ~50–70 vol% of serpentine[57,58]. Antigorite has a highly anisotropic phyllosilicate structure[33], which is very compressible in the cross-plane direction (the [001] direction), and relatively stiff in the in-plane direction[34,36]. Therefore, antigorite is characterized by a weak rheology and, during shear deformation, it likely forms a CPO, in which the [001] direction tends to align perpendicularly to the shear plane/foliation[38,39,59]. In other words, the serpentinized rocks of the slab accommodate most of the shear deformation developed during subduction[60] by orienting antigorite's [001] direction perpendicularly to slab's dip[38,39]. This configuration might create a layer of rocks, where the seismic wave that propagates normal to the plane of slab subduction is slower than in other directions[61].

The combination of antigorite's CPO and strong thermal conductivity anisotropy as revealed by our present results can play a key role in triggering intermediate-depth earthquakes. In particular, the insulating effect of oriented antigorite can promote dehydration embrittlement in the upper plane of a DSZ and potentially thermal runaway within the serpentinized faults (Fig. 7). Dehydration embrittlement is typically expected to be responsible for the upper seismic plane of a DSZ, because the hydrous minerals, e.g., antigorite, contained in the outer ~10 km of the slab are largely exposed to the rising

temperature[9,10]. As shown in our thermal evolution models, the presence of a 3-km-thick thermally-insulating layer of serpentinites ($\varphi_{Atg} > 50\%$, models 9 and 10, which is achieved in old oceanic lithosphere with slow-spreading ridges, e.g., Lesser Antilles[57,58]) efficiently traps the heat at the base of the oceanic crust (~7 km from the slab surface), thus causing a temperature increase of ~50–150 K and an ~7–15 km upward shift of the dehydration front (Table S5). In other words, the antigorite's thermal insulating effect facilitates dehydration embrittlement in the outer 10 km of the slab, while hindering the same mechanism in the inner regions of the slab by preserving it in colder conditions. Such effect is expected to enhance the separation of the two seismic planes of DSZs, and should be taken into account to better understand the features of global DSZs[4].

Moreover, the temperature differences caused by antigorite's thermal insulating effect may contribute to local seismic anomalies. Using the scaling parameter reported by ref. 62, temperature differences of +170 K and −65 K between model 10 and model 1 would lead to a seismically detectable perturbation of compressional velocity $V_p$ of about $-8.2 \times 10^{-2}\,km\,s^{-1}$ and $+3.1 \times 10^{-2}\,km\,s^{-1}$, respectively (i.e., about −1% and +0.4% of the $V_p$ in the upper mantle[63]). In the case of 50% serpentinization (relevant to, e.g., Lesser Antilles[57,58]), temperature differences of +80 K and −25 K would lead to a $V_p$ perturbation of about $-3.8 \times 10^{-2}\,km\,s^{-1}$ and $+1.2 \times 10^{-2}\,km\,s^{-1}$ (about −0.5% and +0.15%, respectively, of the $V_p$ in the upper mantle), which may be difficult to clearly observe by seismology. In addition, the uncertainties in inferring the thermal structure from seismic interpretation are estimated to be about ±100 K in the upper mantle[63], which is comparable to the thermal anomaly produced in our models. Seismic perturbations, however, can also be attributed to several factors other than temperature, including (1) the intrinsic density of a given mineral phase (antigorite is less dense than olivine), (2) the composite elastic properties of the mineral assemblage, and (3) the presence of fluid/melts, etc. Thus, it would be challenging to attribute univocally a seismic or temperature anomaly to a single factor.

On the other hand, the seismicity in the lower plane of a DSZ is more enigmatic and under debate. Depending on the variable temperature profile and water content, etc., in a subduction zone, the lower seismic plane could be caused by either also the dehydration of hydrous minerals[9,13], or other rupture mechanisms. Another plausible trigger for intermediate-depth earthquakes is the thermal runaway[12,19–22], a ductile deformation mechanism in shear zones, where the intense frictional heating induces weakening of rocks and causes the formation of a self-localizing slip planes. To produce such slip planes, the rate of heat production inside the shear zone has to be larger than the rate of heat dissipation from the shear zone[19]. When this condition occurs, the net heat confined inside the shear zone leads to a temperature increase, which promotes ductile deformation and generates weak slip planes. The severe frictional heat sometimes can lead to the formation of lenses of molten rock (e.g., pseudotachylites[20,64]).

Although we have not tested this mechanism in our models, we propose that antigorite's anisotropic thermal insulating effect can promote the onset of thermal runaway. The ideal setting to trigger thermal runaway are the pre-existing shear zones[22], in particular the highly serpentinized normal faults produced in trench-rise system. It has been suggested that mineral transformations, such as dehydration of antigorite, in the fault zones would assist the fault slip that generates heat, leading to self-localizing thermal runaway (see ref. 12 and references therein). During shear deformation, the antigorite crystals hosted within the fault zones should orient themselves with the [001] direction perpendicular to the slip plane. In this configuration, the heat produced by friction can flow easily along the slip plane (due to the high $\Lambda^{010}$), but rarely escapes from the shear zone[60] due to the thermal blanketing effect by the surrounding antigorite's low $\Lambda^{001}$. Thermal runaway is also aided by the weak mechanical resistance of antigorite crystals along the basal planes[17]. Moreover, to achieve runaway

deformation regime, the background temperature of the shear zone has to be lower than a critical temperature $T_c$, below which the thermal relaxation generates unstable ductile deformation[19]. Typically, the critical temperature of a mantle rock is <700 K[65]. Our models show that the thermal insulating effect of oriented antigorite keeps the slab interior in colder temperatures, which may be a necessary condition for the onset of thermal runaway. Potentially, the antigorite-assisted thermal runaway could be responsible for the lower seismic plane of a DSZ[12]. To better our understanding of the thermal runaway mechanism, future numerical modelings on the effect of anisotropic heat diffusion inside shear zones are required to test such scenario.

Finally, the anisotropic thermal conductivity of antigorite can also be relevant in the serpentinized forearc regions[66]. The surface heat flow on top of the forearc is typically very low (~30–40 mW m$^{-2}$) and, in old subduction zones, is uniform from the trench to the volcanic arc. Such anomalous constant value may be associated with the presence of insulating antigorite at depth. Future corner flow models[67] should include antigorite's anisotropic thermal conductivity to compute the rheology of a serpentinized mantle wedge.

In conclusion, we have experimentally demonstrated a strong thermal conductivity anisotropy of antigorite under high P-T conditions along subduction, where the conductivity along the [001] is ~3–4 times lower than that along the [010]. Our 2D numerical simulations show that such anisotropy combined with its CPO during subduction can significantly influence the thermal evolution of a sinking slab with high degrees of serpentinization ($\geq 50\%$). In particular, when antigorite is oriented with the [001] direction perpendicular to the slab dip, it acts as an effective thermal insulator that hinders heat flowing toward the slab interior. This effect increases the temperature at the interface between the lower crust and the lithospheric mantle, thus promoting dehydration embrittlement in the outer ~10 km of the slab. Potentially, the anisotropic insulation by oriented antigorite can promote thermal runaway within the serpentinized faults inside the slab, triggering intermediate-depth earthquakes. Future experimental studies on the thermal conductivity of relevant phyllosilicates in the sediments and crust (e.g., muscovite, biotite, chlorite, phengite, and talc, etc.) along specific crystal orientation and P-T conditions of slab subduction will offer more comprehensive datasets for the thermal properties of slab's minerals. Combined with detailed thermal evolution modeling, these would substantially advance our understanding of important geological, geophysical, and geodynamical features in a subduction zone, including the thermal state, flow dynamics, water transportation, magmatism, and seismic structure of the region.

## Methods

### Sample characterization and preparation

We collected natural antigorite samples from Baibao River (Hualien, Taiwan) and used energy-dispersive X-ray spectroscopy to determine their average chemical composition—$(Mg_{2.80}Fe_{0.05})Si_{2.08}O_5(OH)_{3.77}$ with a density of $2.58 \pm 0.03\,g\,cm^{-3}$. We also performed X-ray diffraction and Raman spectroscopy measurements (Supplementary Fig. S1) to confirm antigorite's crystal structure and vibrational spectrum, respectively, which are both in good agreement with RRUFF standard database for antigorite[68]. We oriented the antigorite crystals to two principal directions along the in-plane b-axis [010] and cross-plane c-axis [001] by either electron backscattered diffraction or single-crystal X-ray diffraction.

For high pressure thermal conductivity measurements at room temperature, we first manually polished the crystals down to a thickness of ~30 μm, and then deposited an ~90-nm-thick aluminum (Al) film on the crystals. We loaded the crystal sample along with a few ruby balls into a symmetric piston-cylinder diamond anvil cell (DAC) equipped with a pair of 500 μm culets and a stainless-steel gasket pre-indented to ~80 μm thick. We compressed the crystal by loading silicone oil as the pressure medium, see Supplementary Fig. S2 for the

schematic illustration of our sample geometry and experimental set-up. We calibrated the pressure within the sample chamber by the shift of the ruby $R_1$ fluorescence peak[69] with an uncertainty of -0.1–0.2 GPa.

To perform simultaneous high P-T thermal conductivity measurements, we used an externally-heated BX-90 DAC (EHDAC), in which a toroidal resistive heater was placed next to the sample chamber, providing a spatially-homogeneous high T environment within the small sample chamber. We compressed the sample by high-pressure gas loading of Ar as the pressure medium. During our high P-T measurements, we also inserted a gas membrane in the EHDAC, allowing us to in situ control the pressure during heating. The pressure uncertainty is -0.1–0.5 GPa due to the broadening of ruby peak upon heating. Details of the EHDAC assemblage and pressure uncertainty during high P-T measurements were described in refs. 27,70.

## Thermal conductivity measurements
We measured the lattice thermal conductivity of single-crystal antigorite at high pressure and a wide range of temperature conditions by time-domain thermoreflectance (TDTR). TDTR is a well-developed ultrafast optical pump-probe method that enables precise measurement of the thermal conductivity of a material[29,71]. In our TDTR setup, we split the output of a mode-locked Ti: sapphire laser into a pump beam, which heated the Al film coated on the antigorite sample, and a probe beam, which monitored the changes in optical reflectivity of the Al film caused by temperature variations. The small intensity changes of the reflected probe beam, synchronous with the 8.7 MHz modulation frequency, were measured by a silicon photodiode detector coupled with a lock-in amplifier. Details of the TDTR were described elsewhere, e.g., ref. 71.

We determined the thermal conductivity of the sample by comparing the ratio of thermoreflectance in-phase signal $V_{in}$ to out-of-phase $V_{out}$ (-$V_{in}/V_{out}$) with calculations based on a bidirectional heat-flow thermal model, see, e.g., refs. 70,72 and references therein. The thermal model requires inputs of several parameters, including laser spot size (7.6 μm), and thickness, thermal conductivity, and volumetric heat capacity of each layer (i.e., silicone oil or Ar, Al film, and antigorite). All input parameters are pre-determined or in situ measured during the measurements (see, e.g., refs. 72–75 for details), while antigorite's thermal conductivity is the only significant unknown and free parameter to be determined. The volumetric heat capacity of antigorite is a key parameter when determining its thermal conductivity. We calculated the P-T dependent volumetric heat capacity of antigorite based on the results by ref. 32 combined with its equation of state[44]. Note that the uncertainty of our thermal conductivity data majorly arises from analysis uncertainty, instead of measurement uncertainty. We estimated the data uncertainty by evaluating the total uncertainties propagated by all the input parameters used in the thermal model, see Supplementary Figs. S3, S4 as well as relevant references therein. We found the error is -10% at P < 5 GPa and <20% at 13 GPa.

## Numerical modeling
We designed a 2D model of slab subduction to investigate the effects of the strong anisotropy of antigorite's thermal conductivity on the heat propagation within the slab. The full description of the physical and numerical model of the slab, its governing equations, the limitations, and the code benchmark are reported in the Supplementary Information (Note S1–S7, Figs. S4–S10, Tables S1–S4, S7). We wrote a MATLAB code to compute the heat diffusion equation $\rho C_p(\partial T/\partial t) = -\nabla \cdot (-\Lambda \nabla T)$ with the finite difference method[76] (Note S2). The symbols represent: $\Lambda (W\,m^{-1}\,K^{-1})$ thermal conductivity (Note S3), $\rho (kg\,m^{-3})$ density (Note S4), $C_p(J\,kg^{-1}\,K^{-1})$ heat capacity (Note S5), $T(K)$ temperature, $t(s)$ time, and $\partial T/\partial t (K\,s^{-1})$ temperature evolution over time. In 2D, the parameter $\nabla T(K\,m^{-1})$ indicates the temperature gradient along the vertical ($\partial T/\partial y$), and the horizontal direction

($\partial T/\partial x$) (Note S2). We simplified the slab as a 600 × 120 km rectangle, and we discretized the domain into a mesh of square cells $\triangle x = \triangle y = 500\,m$ (i.e., 242 × 1202 nodes), see Figs. S5–S7. In the finite difference method[76] (Note S2), $\partial T$ was approximated as the temperature difference between two consecutive nodes ($\triangle T = T_i - T_{i+1}$), and the temperature gradient $\nabla T$ was computed considering the vertical ($\partial y \sim \triangle y$) and horizontal ($\partial x \sim \triangle x$) grid spacing, i.e., $\nabla T_V = \triangle T/\triangle y$ and $\nabla T_H = \triangle T/\triangle x$. For the calculation we set the time step to $\partial t \sim \triangle t = 2000$ yrs. At the four edges of the slab we added an extra stencil of nodes (Fig. S6), in which we set isothermal boundary conditions (Note S2). The slab thickness (120 km) was computed using the analytical solution $2.32\sqrt{\kappa t}$[77], assuming a thermal diffusivity of $\kappa = 1 \times 10^{-6}\,(m^2 s^{-1})$ and a slab age of 80 Myrs[78] (Note S1). The slab was composed of three layers (Fig. S6): (1) a 7-km-thick crust[79], (2) a 3-km-thick layer of hydrous lithosphere[80], and (3) a 110-km-thick dry lithosphere. For simplicity, we assumed the lithospheric slab and the oceanic crust to be composed entirely of olivine, while the hydrous layer was made of an assemblage of antigorite and olivine with variable fractions of $\varphi_{Atg}$ and $\varphi_{Ol} = 1 - \varphi_{Atg}$ (Note S3). The initial temperature profile of the slab at the trench was computed with the half-space cooling equation[77] (Note S1, Equation 2, Fig. S5). In the simulation, the slabs subducted vertically (i.e., dip angle $\theta_{dip} = 90 ♪$), with a constant velocity ($v_{sink} = 5\,cm\,yr^{-1}$) (Note S1, Fig. S7). Subduction occurred from 120 km depth to avoid the interactions between the slab and the overriding plate[81] down to 230 km, i.e., the typical breakdown depth of antigorite[3]. During subduction, the increasing P-T conditions were taken from the adiabatic profile of the mantle[82] (Note S1, Fig. S8), while the P-T-phase evolution of each parameter was computed with the equations extrapolated from literature (Note S3–S5, Fig. S9): (A) olivine $\Lambda^{DryOl}$, $\Lambda^{WetOl41,47}$, $\rho^{Ol83}$ and $Cp^{Ol83}$; (B) antigorite $\Lambda^{Atg}$ (this study), $\rho^{Atg32,44}$, $Cp^{Atg32}$. The aggregate thermal conductivity of the hydrous layer was computed with the geometric average of olivine and antigorite contributions[84] (Note S3): $\Lambda^{HydLayer} = (\Lambda^{Ol})^{\varphi Ol*}(\Lambda^{Atg})^{\varphi Atg}$. We chose to use geometric averaging because it provides the best correlation between the computed and the measured thermal conductivity of a random grainy medium[85]. To investigate the anisotropy of antigorite's thermal conductivity we distinguished the horizontal $\Lambda_H$ and vertical $\Lambda_V$ components along the corresponding axis (Note S2). The limitations of the model (Note S6) include: (1) the 2D geometry, which produces a colder slab compared to the 3D geometry; (2) the simplified petrology, composed only by olivine and antigorite; (3) the P-T independent mineral assemblage; (4) the absence of self-consistent gravity-driven subduction; (5) the lack of additional heating source (e.g., radioactive contribution); (6) the limited domain size (the surrounding mantle is only 500 m thick); (7) the lack of interactions between the slab and the overriding plate; (8) the absence of interactions between the slab and the mantle wedge (corner flow).

## Data availability
The main data generated in this study are available in https://zenodo.org/records/11104058.

## Code availability
The MATLAB script used to perform numerical modeling is available in https://zenodo.org/records/11104058.

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

## Acknowledgements

This work was partially supported by the Academia Sinica and the National Science and Technology Council (NSTC) of Taiwan, Republic of China, under Contract IA-111-M02 (to W.P.H.) and 110-2628-M-001-001-MY3 (to W.P.H.). Y.H.C. acknowledges support by the NSTC 105-2922-I-008-212. E.M. acknowledges the support of the Helmholtz Young Investigators Group CLEAR (VH-NG-1325) and DFG project - BL 1690/1-1. The numerical model was written in MATLAB using the license 139702 (GFZ_network_concurrent). E.M. would like to thank N. Satta, and G. Criniti for the fruitful discussion, M. Jarema for the support in optimizing the MATLAB script, and S. Sarkar for English proof reading.

## Author contributions

W.P.H. conceived, designed, and supervised the project. Y.H.C., Y.C.T., and W.P.H. conducted experiments and analyzed data. E.M. performed numerical modeling. All authors wrote, reviewed, and commented on the manuscript.

## Competing interests

The authors declare no competing interests.
