## [Peer Review File · Nature Communications]

Anisotropic thermal conductivity of antigorite along slab subduction impacts seismicity of intermediate-depth earthquakesReviewers' comments:

Reviewer #1 (Remarks to the Author):

The paper discusses the role of anisotropic thermal conductivity of serpentine in producing the double seismic zone that is seen in many subducting slabs. The experimental data are novel because they constrain the anisotropy of the thermal conductivity at elevated pressure of the first time, and potentially important because the anisotropy is found to be very large. The authors discuss possible implications for earthquake generation with the use of thermal models of the subducting slab. They draw interesting conclusions, particularly about the nature of the lower part of the double seismic zone as related to insulation of the slab due to the anisotropic thermal conductivity. The paper should be published after addressing the following comments:

1) Fig. 1. The data show that the anisotropy initially increases with increasing pressure. This is somewhat surprising because other experiments show that the structure becomes less anisotropic with increasing pressure (c/a and c_{22}/c_{33} both decrease with increasing P). This feature of the data should be discussed.

2) Fig. 1. The comparison with the polycrystalline data is somewhat puzzling. One might expect the current measurements to bracket the polycrystalline result, but they do not: the polycrystalline result is nearly coincident, or even smaller than the [001] direction over much of the pressure range. The authors should discuss the reason for this.

3) The numerical model is in the end quite simple: simply replacing one layer in the standard treatment with aligned antigorite. It is therefore surprising and a bit annoying that the model and modeling results are discussed in such a lengthy and somewhat convoluted way. Any effort to simplify and clarify this discussion, including perhaps a reduction in the number of somewhat redundant figures (e.g. 3/4, 5/6), would be appreciated.

4) The numerical model is of course, by necessity, over-simplified, but there is at least one aspect that requires further discussion. The authors assume perfect alignment of serpentine crystals within the anisotropic layer. This seems to be a very strong assumption that requires justification, either on the grounds of experimental results on texture development and the likely stress state in the slab, and/or on the grounds of in situ observations of seismic anisotropy.

Reviewer #2 (Remarks to the Author):

Although the paper is an elaborated work, but less impact to the community. It is appreciable to experimentally demonstrate the anisotropy of thermal conductivity of antigorite. However, there are

only limited descriptions of starting samples and sample preparation, which making difficult to evaluate the reliability of the experiment. Antigorite has a characteristic crystal structure. It is critical to discuss the anisotropy based on the crystal structure.

Most of this paper is devoted to the temperature analysis in the slab. It may be reasonable that the thermal conductivity of antigorite relates to the double seismic surface. It may be reasonable as well that the upper seismic surface is caused by the dehydration of antigorite. However, it is difficult to immediately accept that the lower seismic surface is caused by the thermal runaway. Thermal runaway increase the temperature at the bottom of the slab, but it is unclear why it causes local earthquakes at the specific position of the lower seismic surface. The argument is far from logical.

Here are the reviewer's impressions. This paper should be divided into two parts. One is an experimental paper, in which characteristics of sample should be described well and the relationship between the thermal conductivity anisotropy and antigorite crystal structure should be discussed. The other is the slab temperature simulation. Please do not explain everything. The authors should focus only on phenomena with higher probability.

Reviewer #3 (Remarks to the Author):

This manuscript presents experimental measurements of the anisotropy of thermal conductivity in single crystals of serpentine (antigorite), under high pressure and high temperature. The authors then use numerical modeling to assess the consequences of this anisotropy on the thermal state of (cold) subducting slabs.

The same strategy has been used by some of the authors in at least one previous publication on the hydrous phase D in 2022, in *Journal of Geophysical Research*. The present manuscript and the supplementary material follow a very similar outline as the 2022 publication. Therefore, the manuscript does not report a novel kind of experiment or model. The application to serpentines in a subduction zones context is, however, new to my knowledge.

The data and numerical models obviously are of great interest to the communities of mineral physics and modeling, and deserve publication. However, I believe the authors should first address some major points that follow on the form (1 to 3) and on the science (4 to 9). One of these points is the experimental measurements seem to be incomplete as the a-axis thermal conductivity was not measured (see comment 4 below). A second one is, the discussion requires a major reworking and strengthening. There is a wide gap between the numerical model part which remains descriptive, and the discussion itself which is purely speculative and superficial, "pasting" the results on broad issues currently raised in the literature. Because of these two weaknesses, and because the strategy and methods are not novel, I am not 100% convinced of the necessity to publish these results in *Nature communications*.

The manuscript is correctly written overall, but I do have a couple of major comments on the form.

1) I understand space is limited for manuscripts in this journal. However, I think the choices made in the distribution of figures and text information do not work so well in the present version, and the referencing between the main and supplementary text should be done properly.

For instance:

- between lines 104 and results, I do not find information on the experiments. The details are the purpose of the methods section, but the reader still needs a minimum amount of information to understand the experimental strategy in the main text. There should be an explicit reference to the method section at this stage, too.

- the setup for the numerical model is unclear from the main manuscript and figure 4, cannot be understood from the main text, or without referring to the suppl. material. There should have been references to the supplementary information at appropriate places (eg. line 189 there should be a reference to table S4, line 213 to figure S5, etc.).

- the limitations of the model are presented in the supplementary text only, while the biggest ones raise points that should be in the main discussion.

2) The references to the literature, in my view, tend to favor some authors or a couple of (recent) visible papers - not always the most relevant, among a much wider literature, and does not make a fair share to experimental works outside of a few references (this is especially true in the introduction).

3) The description of the results from the numerical part is heavy to read and should be simplified (and this would leave space for instance for an in-depth discussion).

Then I have some major reservations on the scientific part:

4) The authors measured single crystal thermal conductivity along two crystal directions b and c. Why has the a axis not been measured ? As far as I saw, there is no justification and this omission is odd. The crystal structure is different along the a axis than along the b (and c) axes, with reversals of tetrahedral layers that may affect thermal conductivity as well.

5) In the absence of X-ray diffraction to confirm in-situ that the orientation of the single crystal remains stable during compression, and the measurements, is the orientation of the sample in the diamond anvil cell well controlled? how does this contribute to the uncertainty of the thermal conductivity measurement along a specific crystal orientation?

6) There is no discussion on the possible role of grain boundaries, or the applicability of the single crystal measurements of conductivity to rocks, which are polycrystals. This should be addressed somewhere.

7) A texture with 100% of crystals oriented with c axis normal to the slab surface is an extreme case. What would be the thermal conductivity of a rock with an intermediate, more realistic, fabric? The effect of CPO is tested here only on two extreme unrealistic cases. This, in my view, does not allow writing the sentence line 363.

8) The discussion is only speculative. There is a mix up with the scenario of thermal runaway and melting, lines 370 to 375. Then, lines 376 to 396, the thermal runaway scenario proposed does not explain clearly how antigorite is involved. The ref. 49 is based on a specific olivine rheology, which is different from antigorite rheology, and from its frictional properties. From figure 8, the thermal runaway seems to occur in antigorite itself (?) (to my knowledge has not been tested numerically). Figure 8 should be clarified accordingly.

9) In the case such thermal distributions would be real, I wonder whether they should also be seen in seismic velocities or lead to a re-interpretation of some seismic observations?

Less major points:

10) It is interesting, but is not made clear: why testing the influence of thermal conductivity of hydrous olivine (lines 283-290) - if we take the starting point of the manuscript, which is the investigation of the effect of the anisotropic thermal conductivity of antigorite.

11) In the methods section line 453-454, the reader is only referred to previous publications for P and T uncertainties, and there are no error bars, or information on the uncertainty, on figure 1 and 2 for the pressure (and temperature fig. 2). For general readers, this could be the first paper they read, using this experimental technique. Please provide some information in the main text or in the methods section, on the uncertainties for P.

12) Figure 7 could have arrows hotter/colder for instance, to facilitate understanding as first sight.

Line by line comments:

42: ref. 4, I believe there are anterior references for this.

Introduction lines 46 from 53: the publications listed here are mostly from non-experimental work, while the literature is abundant on this topic. Experimental works actually played a very important role, if not the most important, in proposing all these mechanisms.

Line 51: Reynard, B., J. Nakajima, and H. Kawakatsu (2010), Earthquakes and plastic deformation of anhydrous slab mantle in double Wadati-Benioff zones, *Geophys. Res. Lett.*, 37, L24309, doi:10.1029/2010GL045494 may be cited also here.

Line 56, ref 20 unnecessary, while many other would be relevant.

Line 63 : ref. 7 could also be cited here.

Line 128: ref. 62 could also be cited here.

182-183: there should be a reference to the supplementary material for more information.

Lines 213-215: I do not understand the connection between the first and second part of this

sentence (before and after “hereafter”).

Lines 286 : I am not sure what ‘similarly’ refers to (did you mean the differences between wet and dry are similar to the differences between dry olivine and models that include antigorite ?)

Line 304: “moreover” and “further” is kind of repeating

Line 304-305 : I find “antigorite stability field” clumsy here, because for mineral physicists, stability field refers to a P-T stability field in the thermodynamic sense, a property of the mineral which will not vary - “breakdown depths”, as used later in the text, is more appropriate to me.

Lines 349 to 353: this scenario is presented like a certainty with only one reference, while I believe it is debated.

Lines 366-368 there is only one reference here - it would be good to have additional references, including modeling and mineral physics ones.

Conclusion, line 425: I believe the influence of DHMS will depend on their amount at depths.

Figure S5: the single-ended arrows: what do they mean?

In what follows, all reviewers' comments are in italics and our point-to-point responses are in normal fonts with blue color. Changes in our revised manuscript are also labeled by blue color. We have added a significant amount of materials and information and revised our discussions, which have fully addressed the reviewers' concerns and questions.

Report of Referee # 1

1. Fig. 1. The data show that the anisotropy initially increases with increasing pressure. this is somewhat surprising because other experiments show that the structure becomes less anisotropic with increasing pressure (c/a and c_{22}/c_{33} both decrease with increasing P). This feature of the data should be discussed.

We are afraid that the reviewer may have misread our data in Fig. 1. Our data indeed show that the thermal conductivity anisotropy ($\Lambda^{010}/\Lambda^{001}$) at ambient conditions is ~ 4.3 and decreases to ~ 3.1 at $P \sim 2$ GPa, to ~ 2.5 at $P \sim 4$ to 5 GPa. From $P \sim 5$ to 7 GPa, the anisotropy slightly increases from ~ 2.5 to ~ 3.7 . Thus, upon initial compression, the thermal conductivity does become less anisotropic. However, as shown in the Fig. 1 of Satta et al. GRL 49, e2022GL099411(2022) (Ref 35), at $P > 6$ GPa, compared to their original trends, the in-plane elastic constants seem to increase, while the cross-plane elastic constant decreases. Such changes lead to an increase in the elastic anisotropy at $P > 6$ GPa, in good agreement with the trend in thermal conductivity anisotropy we observed. (This comment is a bit similar to the comment #1 of reviewer #2 below, so please also see our reply there for more details.)

To point out the feature of thermal conductivity anisotropy and discuss the mechanism, in line 122-129, we have added the following sentences:

“In other words, the thermal conductivity anisotropy ($\Lambda^{010}/\Lambda^{001}$) at ambient conditions is ~ 4.3 and becomes less anisotropic as the anisotropy progressively decreases to ~ 2.5 at $P \sim 4$ to 5 GPa; afterwards the anisotropy slightly increases to ~ 3.7 at 7 GPa. Since thermal conductivity is an ensemble contribution of heat transport that involves heat capacity, elastic constant, and phonon mean-free-path of all the available phonon modes³⁶, such trend with pressure can be primarily accounted for by the pressure evolution of the elastic constant along the in-plane and cross-plane direction, see, e.g., Ref ³⁵.”

2. Fig. 1. The comparison with the polycrystalline data is somewhat puzzling. One might expect the current measurements to bracket the polycrystalline result, but they do not: the

polycrystalline result is nearly coincident, or even smaller than the [001] direction over much of the pressure range. The authors should discuss the reason for this.

The literature data for polycrystalline antigorite by Horai JGR 1971 and Osako *et al.* PEPI 2010 are indeed bracketed by our *b*- and *c*-axis data at ambient and $P < 4$ GPa, while Osako *et al.*'s data become comparable to our *c*-axis data at $P \sim 4$ to 9 GPa. In fact, Osako *et al.*'s data appear to be a constant, or at least, very weakly dependent on the pressure. Such response to pressure is rare, at odd with a typical trend that most materials' thermal conductivity increases with pressure. We are not aware of the detailed experimental conditions in the study by Osako *et al.*, and thus could only suspect or propose that their polycrystalline sample may have been preferentially oriented to near the *c*-axis under the compression conditions in their experiments. By contrast, as we addressed in comment #9 of reviewer #3, our crystal orientation remains essentially the same throughout our measurements (see our explanations there for more details).

To discuss the comparison of our present data with literature data, we have added the following paragraph in line 138-152:

“In Fig. 1 we also plotted literature results for polycrystalline antigorite at ambient pressure³⁰ (open blue circle) and to 8.5 GPa³¹ (open green stars), respectively, which are bracketed by our *b*- and *c*-axis data at $P < 4$ GPa. Nevertheless, the data by Osako *et al.*³¹ (open green stars) appear to be nearly a constant, rather than a typically increasing trend upon compression, and become comparable to our *c*-axis data at $P \sim 4$ to 9 GPa. We are not aware of the detailed experimental conditions in Osako's study, and thus could only propose that their polycrystalline sample may have been progressively oriented preferentially to near the *c*-axis under their compression conditions. By contrast, to check the potential variation of crystal orientation during our compression conditions, we have re-characterized the orientation from the quenched samples out of the DAC using electron backscattered diffraction or X-ray diffraction (*in situ* measurements of crystal orientation and thermal conductivity at high pressures are currently not available). We found that the orientation remained essentially the same, suggesting that the minor variation, if there is, of crystal orientation is expected to have minor effects on the uncertainty of our antigorite's thermal conductivity.”

3. *The numerical model is in the end quite simple: simply replacing one layer in the standard treatment with aligned antigorite. It is therefore surprising and a bit annoying that the model and modeling results are discussed in such a lengthy and somewhat convoluted way. Any effort to simplify and clarify this discussion, including perhaps a reduction in the number of somewhat redundant figures (e.g. 3/4, 5/6), would be appreciated.*

Following the reviewer's suggestion, we have substantially simplified the descriptions of numerical modelling and results. Now the section is easier to read and it flows better. Moreover, we have moved the original Fig. 5 (heat flux Q) to the Supplementary Information (now becomes Fig. S11). We believe that both Figs. 3 and 4 are essential. Figure 3 is important to visualize and understand our model design, and Fig. 4 illustrates the final outcome of the models (i.e., the numerical results). Although the model is simple, its significant outcome demonstrates the large-scale implications of antigorite's anisotropic thermal conductivity.

4. The numerical model is of course, by necessity, over-simplified, but there is at least one aspect that requires further discussion. The authors assume perfect alignment of serpentine crystals within the anisotropic layer. This seems to be a very strong assumption that requires justification, either on the grounds of experimental results on texture development and the likely stress state in the slab, and/or on the grounds of in situ observations of seismic anisotropy.

Several field observations, deformation experiments, and theoretical calculations have reported that antigorite has a strong crystal preferred orientation (CPO), in which its [001] direction tends to orient perpendicularly to the shear direction. For example, Padrón-Navarta et al. *EPSL* 2012 (Ref 38) reported a strong foliation in natural serpentinites samples (Cerro del Almirez, Spain): antigorite's [001] shows a peak of relative density of 4.9-17.4 m.u.d. (multiples of uniform distribution) along the Z-axis, which is perpendicular to the shear direction (X-axis). Moreover, the majority of antigorite grains orient their [001] direction $\pm 30^\circ$ from the Z-axis. Similar behaviors have also been reported by Nishii et al., *Journal of Structural Geology* 33, 1436 (2011) (4.7-18 m.u.d.) and Jung, *EPSL* 307, 535 (2011) (5 m.u.d.). On these bases, it is reasonable to assume that, during subduction, antigorite would orient its [001] direction normal to the main shear direction, i.e., along the slab dip.

In the model we wanted to test the ideal case where all antigorite crystals are oriented with their [001] normal to the slab dip (i.e., the horizontal insulator scenario, model set 3 in our manuscript). We considered this ideal case as one end-member, and we also modelled a completely opposite case, where all antigorite crystals are oriented with their [001] parallel to the slab dip (i.e., the vertical insulator scenario, model set 2). Our simplified approach may not necessarily reflect realistic antigorite orientation inside the slab, while our model aimed to test these two end-member cases to study the large-scale effects of antigorite's anisotropic thermal conductivity on heat diffusion inside the slab.

Unfortunately, it is not possible to find ophiolites of subducted slab which maintained the CPO gained during subduction. Moreover, to our knowledge, it is not yet

possible to infer the orientation of antigorite inside the slab from seismology. Satta et al., GRL 2022 (Ref 35) reported that in order to see 1 second delay on the shear-velocity, the seismic wave has to cross a >10 km-long path of fully serpentinized rocks (100% alteration), and the incident angle of the seismic wave must be parallel to the foliation plane. Given such conditions and the limited time sensitivity of modern seismometers (~ 10 milliseconds), it is almost impossible to tell the orientation of a partially serpentinized rock layer (<70% alteration) (see Allen et al. 2022, Ref 55, and Cooper et al. 2020, Ref 56) which is crossed by a non-ideally oriented seismic wave for less than 10 km. Therefore, we can only make an educated guess and assumed that antigorite [001] direction is oriented perpendicularly to the slab dip within $\pm 30^\circ$.

Finally, we argue that our model assumption of perfect crystal alignments along two principal axes is reasonable. Our TDTR measurements report the upper and lower bounds of antigorite's thermal conductivity depending on the orientation of the crystal. We can define φ as the incident angle between the heat flux and the foliation plane of antigorite. On one hand, if the [001] direction is parallel to the heat flux (i.e., perpendicular to the {001} foliation planes $\varphi = 90^\circ$), the heat will flow slowly because of the low Λ^{001} . On the other hand, if the [001] direction is perpendicular to the heat flux (i.e., parallel to the {001} foliation planes $\varphi = 0^\circ$), the heat will flow faster because of the high Λ^{010} . If the antigorite is oriented with an incident angle between $0^\circ < \varphi < 180^\circ$ and $\varphi \neq 90^\circ$, we can assume that the thermal conductivity would be a value between Λ^{001} and Λ^{010} . In this case, we can use a simple function: $\Lambda_{Atg}(\varphi) = (\Lambda^{001})^{\sin\varphi} * (\Lambda^{010})^{1-\sin\varphi}$.

Here we used a geometrical average because Λ^{001} is much lower than Λ^{010} , and hence it will act as a bottleneck to the lattice heat transport (Grose & Afonso, 2019). With this equation we can calculate antigorite's thermal conductivity as a function of its orientation with respect to the heat flux, e.g., at ambient pressure:

φ	90 °	80 °	60 °	45 °	30 °	10 °	0 °
$\Lambda_{Atg} [W m^{-1} K^{-1}]$	1.14	1.16	1.38	1.72	2.31	3.68	4.71
φ	90 °	100 °	120 °	135 °	150 °	170 °	180 °
$\Lambda_{Atg}[W m^{-1} K^{-1}]$	1.14	1.16	1.38	1.72	2.31	3.68	4.71

We believe that this equation is a fair approximation of the thermal conductivity of antigorite when $0^\circ \leq \varphi \leq 180^\circ$: Λ_{Atg} remains low for high incidence angle ($45^\circ \leq \varphi \leq 135^\circ$) because heat would flow through several foliation planes. However, Λ_{Atg} is high when the incidence angle is low ($\varphi \leq 30^\circ$ or $\varphi \geq 150^\circ$) because heat would flow on the foliation planes.

Heat flux is always oriented along the direction of the maximum temperature gradient, which, in our case, is perpendicular to the slab surface. Therefore, we can assume that the

incident angle between heat flux and antigorite's foliation planes is high: $\varphi = 90^\circ \pm 30^\circ$. This range of orientations does not show a major difference in antigorite's thermal conductivity: ~20% at ambient pressure which becomes less than 10% at high pressure (see the plateau of Λ^{001} in Fig. 1 of the main manuscript). Even at $\varphi = 30^\circ$ and $\varphi = 120^\circ$ the $\Lambda^{010}/\Lambda^{001}$ anisotropy remain large: ~3.4 at ambient pressure, and ~2.3 at $P=7$ GPa. For these reasons, we believe that the assumptions made to design our simplified model of heat flow through an anisotropic medium are sufficient to describe the physics of the phenomenon, and our conclusions are still valid even if antigorite is not perfectly aligned.

To justify our assumption of perfect alignment of single-crystalline antigorite along the slab dip angle, we have added a new section in Supplementary Information Text S8, Table S8, and a new figure (Fig. S13).

Report of Referee # 2

1. Although the paper is an elaborated work, but less impact to the community. It is appreciable to experimentally demonstrate the anisotropy of thermal conductivity of antigorite. However, there are only limited descriptions of starting samples and sample preparation, which making difficult to evaluate the reliability of the experiment. Antigorite has a characteristic crystal structure. It is critical to discuss the anisotropy based on the crystal structure.

Our work represents the first direct measurements demonstrating antigorite's strong anisotropic thermal conductivity. Our numerical modelling further indicated that antigorite's thermal insulating effects critically impact slab's thermal evolution and occurrence of intermediate-depth earthquakes. We believe our work brings significant conceptual advances in understanding an important topic (mechanisms for intermediate-depth earthquakes) for the geophysics community.

To provide more information on the starting sample, we have added data for X-ray diffraction and Raman spectrum of our natural antigorite as the new Supplementary Fig. S1. In line 489-495, we have also added and revised the following sentences for more descriptions on the characteristics of starting samples, including our X-ray diffraction and Raman spectroscopy that confirm antigorite's crystal structure, as these results are both in good agreement with RRUFF standard database for antigorite:

“We also performed X-ray diffraction and Raman spectroscopy measurements (Supplementary Fig. S1) to confirm antigorite's crystal structure and vibrational spectrum, respectively, which are both in good agreement with RRUFF standard database for antigorite⁶⁵. We oriented the antigorite crystals to two principal directions along the in-

plane *b*-axis [010] and cross-plane *c*-axis [001] by either electron backscattered diffraction or single-crystal X-ray diffraction.”

For the sample preparation, we think we have provided enough information and references so that the readers can follow and reproduce the experiments.

Regarding the discussion on the thermal conductivity anisotropy based on antigorite’s layered crystal structure, we actually mentioned this point in the Introduction line 81-85: “Given its layered crystal structure, antigorite is characterized by a strong anisotropy in its elastic constants along different crystal orientations³²⁻³⁵. Since the lattice heat transport scales with the elastic constants³⁶, antigorite’s thermal conductivity is expected to be anisotropic as well. Such hypothesis, however, has not been experimentally tested.”

To further echo this point, in line 122-129, we have added the following sentences: “In other words, the thermal conductivity anisotropy ($\Lambda^{010}/\Lambda^{001}$) at ambient conditions is ~ 4.3 and becomes less anisotropic as the anisotropy progressively decreases to ~ 2.5 at $P \sim 4$ to 5 GPa; afterwards the anisotropy slightly increases to ~ 3.7 at 7 GPa. Since thermal conductivity is an ensemble contribution of heat transport that involves heat capacity, elastic constant, and phonon mean-free-path of all the available phonon modes³⁶, such trend with pressure can be primarily accounted for by the pressure evolution of the elastic constant along the in-plane and cross-plane direction, see, e.g., Ref³⁵.” (This comment is a bit similar to the comment #1 of reviewer #1 above, so please also see our reply there for more details.)

2. Most of this paper is devoted to the temperature analysis in the slab. It may be reasonable that the thermal conductivity of antigorite relates to the double seismic surface. It may be reasonable as well that the upper seismic surface is caused by the dehydration of antigorite. However, it is difficult to immediately accept that the lower seismic surface is caused by the thermal runaway. Thermal runaway increases the temperature at the bottom of the slab, but it is unclear why it causes local earthquakes at the specific position of the lower seismic surface. The argument is far from logical.

We understand the reviewer’s concern about the application of the thermal runaway to the lower plane seismicity. We have toned down our discussion by adding “Ferrand, Lithos 2019” as Ref 12 and revising the following sentences in line 416-422 to:

“On the other hand, the seismicity in the lower plane of a DSZ is more enigmatic and under debate. Depending on the variable temperature profile and water content, etc., in a subduction zone, the lower seismic plane could be caused by either also the dehydration of hydrous minerals⁹, or other rupture mechanisms. Another plausible trigger for

intermediate-depth earthquakes is the thermal runaway^{12,18–21}, a ductile deformation mechanism in shear zones, where the intense frictional heating induces weakening of rocks and causes the formation of a self-localizing slip planes.”

We now just emphasize the thermal insulating effect by antigorite’s strong anisotropic thermal conductivity and CPO can promote the thermal runaway, which can trigger an intermediate-depth earthquake (and potentially the lower seismic plane of a DSZ, as suggested by Ferrand Lithos 2019, Ref 12, but the exact seismicity depends on the local stress and many other factors).

3. Here are the reviewer's impressions. This paper should be divided into two parts. One is an experimental paper, in which characteristics of sample should be described well and the relationship between the thermal conductivity anisotropy and antigorite crystal structure should be discussed. The other is the slab temperature simulation. Please do not explain everything. The authors should focus only on phenomena with higher probability.

We believe that the strength of this paper is the multidisciplinary approach of combining experimental measurements of thermal conductivity with the numerical models of slab’s thermal evolution. Moreover, we compared our results with the geophysical observations of intermediate-depth seismicity. Our simple MATLAB script can be easily modified to implement new thermal conductivity data, and our model design has the advantage of being fast to compute (few hours). More sophisticated codes, on the other hand, requires careful implementation of every new parameter (to avoid conflicts between several functions) and the models can take several days to compute. The simplicity of our code is a limit, but also an advantage: we can describe only one physical phenomenon (heat diffusion), but we can also quickly test if the measured thermal conductivity of a key mantle mineral has the potential to alter the geodynamic behavior of the slab. In case it does, we then encourage the numerical modelling community to implement the new thermal conductivity data in their codes to simulate the full thermo-mechanical behavior of the slab.

Splitting the paper in two will not have the same strength: on one hand, the experimental paper will not have the calculations to support the discussion on the large-scale thermal evolution implications; on the other hand, the model presented in the numerical paper will be too simple to be representative of a realistic geological scenario. Together, however, the two approaches complement and strengthen each other. They better emphasize the importance of our finding: we measured experimentally that the antigorite’s thermal conductivity is strongly anisotropic, and we also showed numerically that, on the long term, this anisotropy can affect the thermal structure of the slab with potential implications on its rheology and the breakdown depth of the hydrous minerals.

Nevertheless, as the reviewer suggested, in our revised manuscript we balanced better the experimental and numerical sections (please also see our reply to comment #3 of reviewer #1). We also substantially revised our geophysical discussions to highlight the impacts of antigorite's thermal insulating effect on the different triggering mechanisms responsible for intermediate-depth earthquakes. Finally, we added X-ray diffraction and Raman data (see Figure S1 in the Supplementary Information) to confirm antigorite's orientation and crystal structure, and to describe the correlation between thermal conductivity anisotropy and crystal structure (as we replied in comment #1).

Report of Referee # 3

1. The same strategy has been used by some of the authors in at least one previous publication on the hydrous phase D in 2022, in Journal of Geophysical Research. The present manuscript and the supplementary material follow a very similar outline as the 2022 publication. Therefore, the manuscript does not report a novel kind of experiment or model. The application to serpentines in a subduction zones context is, however, new to my knowledge.

In fact, our present numerical modelling is specifically designed for two-dimensional thermal evolution, which is fundamental to study anisotropy. Here the thermodynamic parameters in a given node are re-calculated after each time step to take into account the non-linear effect of temperature changes. Such strategy and numerical model are **novel and advanced**, compared to our previous studies. Most importantly, we believe that although a new methodology is helpful, it is not necessarily a must-have. **What really matters is whether a significant conceptual advance is presented.** (The groundbreaking experimental discovery of post-perovskite in 2004 was made by an old strategy and traditional method: synchrotron X-ray diffraction within a laser-heated diamond anvil cell. There are a lot more examples where colleagues used such the same approach to create impactful results.) **We believe our combined experimental and numerical findings do present significant conceptual advances in revealing the critical impacts of antigorite on slab's thermal evolution and occurrence of intermediate-depth earthquakes.** We appreciate that the reviewer also recognized such novelty in revealing scientific impacts.

2. The data and numerical models obviously are of great interest to the communities of mineral physics and modeling, and deserve publication. However, I believe the authors should first address some major points that follow on the form (1 to 3) and on the science (4 to 9). One of these points is the experimental measurements seem to be incomplete as the a-axis thermal conductivity was not measured (see comment 4 below). A second one

is, the discussion requires a major reworking and strengthening. There is a wide gap between the numerical model part which remains descriptive, and the discussion itself which is purely speculative and superficial, “pasting” the results on broad issues currently raised in the literature. Because of these two weaknesses, and because the strategy and methods are not novel, I am not 100% convinced of the necessity to publish these results in Nature communications.

We thank the reviewer for recognizing the importance of our present work. We have adequately addressed all your comments or questions as below.

The manuscript is correctly written overall, but I do have a couple of major comments on the form. I understand space is limited for manuscripts in this journal. However, I think the choices made in the distribution of figures and text information do not work so well in the present version, and the referencing between the main and supplementary text should be done properly.

For instance:

3. - between lines 104 and results, I do not find information on the experiments. The details are the purpose of the methods section, but the reader still needs a minimum amount of information to understand the experimental strategy in the main text. There should be an explicit reference to the method section at this stage, too.

To provide basic information about the experiments, we have added the following introductory sentences at the beginning of the results section in line 111-115:

“We used ultrafast time-domain thermoreflectance (TDTR) coupled with DAC to precisely measure the lattice thermal conductivity of single-crystalline natural antigorite to ~13 GPa at room temperature. The TDTR is an ultrafast pump-probe spectroscopy for high-precision thermal conductivity measurements at high pressures and wide range of temperatures, see, e.g., Ref ^{26,29,39} and Methods for details.”

4. - the setup for the numerical model is unclear from the main manuscript and figure 4, cannot be understood from the main text, or without referring to the suppl. material. There should have been references to the supplementary information at appropriate places (eg. line 189 there should be a reference to table S4, line 213 to figure S5, etc.).

To make the descriptions of the numerical modelling clearer, we have added the following sentence in line 217-220:

“The full description of the model is reported in Supplementary Information: physical and numerical model (Text S1-S2, Fig. S4-S8, Table S1), thermodynamic parameters

(Text S3-S5, Fig. S9, Tables S2-S3), model limitations (Text S6), and code benchmark (Text S7, Fig. S10-S11, Table S7).

Please note that all Supplementary Text, Figures and Tables are now explicitly mentioned in the main manuscript. Moreover, in the numerical method section, we have referenced each item specifically.

5. - the limitations of the model are presented in the supplementary text only, while the biggest ones raise points that should be in the main discussion.

To clarify the limitation of our model in the main manuscript, as we replied in the above comment, we have added a brief introductory statement about the modeling and its limitation in line 219 in the main text.

We have also added the following sentence in line 587-594 in Methods:

“The limitations of the model (Text S6) include: 1) the 2D geometry, which produces a colder slab compared to the 3D geometry; 2) the simplified petrology, composed only by olivine and antigorite; 3) the *P-T* independent mineral assemblage; 4) the absence of self-consistent gravity-driven subduction; 5) the lack of additional heating source (e.g., radioactive contribution); 6) the limited domain size (the surrounding mantle is only 500 m thick); 7) the lack of interactions between the slab and the overriding plate; 8) the absence of interactions between the slab and the mantle wedge (corner flow).”

6. The references to the literature, in my view, tend to favor some authors or a couple of (recent) visible papers - not always the most relevant, among a much wider literature, and does not make a fair share to experimental works outside of a few references (this is especially true in the introduction).

In the Introduction, we cited 38 relevant papers as our references, which, we think, cover a broad range of different topics, author groups, and experimental methods (including 1 book, 4 reviews, 8 numerical modelling papers, 6 geophysical observations paper, 2 field observation papers, 17 experimental papers). Given the multidisciplinary approaches used in this study we had to spread the citations among different fields in order to support our description of: 1) the occurrence of intermediate-depth seismicity; and 2) the different triggering mechanisms; 3) the importance of antigorite in the slab; 4) its elastic properties; 5) its propensity to form a CPO; 6) the importance of conductivity measurements to compute the thermal evolution of the slab; 7) why is fundamental to know slab temperature to unravel intermediate-depth seismicity; 8) the lack of thermal conductivity measurements of key mantle minerals, etc. We also believed that, for each topic, we cited state-of-the-art papers (e.g., Ferrand et al., 2016, Nat. Comm.) as well as

pioneering papers (e.g., Hobbs & Ord, 1988, JGR). In the rest of the manuscript we mainly cited papers to thoroughly describe the experimental setup and the model design. Nevertheless, we are happy to include any recommended paper suggested by the reviewer if it represents the state-of-the-art and is better than the works we selected.

7. The description of the results from the numerical part is heavy to read and should be simplified (and this would leave space for instance for an in-depth discussion).

To simplify the description of numerical results, we have substantially revised the section. Now the section is easier to read with better organization and flow. Moreover, we have moved the original Fig. 5 (heat flux Q) to the Supplementary Information (now Fig. S11). We also expanded the discussion section by adding a lot more information and explanations.

Then I have some major reservations on the scientific part:

8. The authors measured single crystal thermal conductivity along two crystal directions b - and c -. Why has the a - axis not been measured? As far as I saw, there is no justification and this omission is odd. The crystal structure is different along the a - axis than along the b - (and c -) axes, with reversals of tetrahedral layers that may affect thermal conductivity as well.

We did measure the a -axis thermal conductivity at ambient conditions, $\sim 4.5 \text{ W m}^{-1} \text{ K}^{-1}$, which is close to the b -axis we presented. Since thermal conductivity scales with the elasticity (square of sound velocity), such behavior can be understood by the similar elastic constant and compressional velocity along these two (a - and b -) axes, see, e.g., Marquardt et al., *Am. Mineral.* 100, 1932 (2015) (Ref 33), Bezacier et al., *JGR*, 118, 527 (2013) (Ref 45), and Satta et al. *GRL* 49, e2022GL099411(2022) (Ref 35). We thus expect that the thermal conductivity along these two orientations in the *in-plane* direction will be similar to each other and much larger than the c -axis. What critically impacts slab's thermal evolution is the thermal insulating effect along the c -axis, not the a - and b -axis; we thus only measured the b -axis thermal conductivity at high pressure-temperature conditions. Lack of a -axis data does not affect our findings.

To clarify this point, we have added the following paragraph in line 153-162:

“Note that we have also measured the thermal conductivity along the a -axis, Λ^{100} , at ambient conditions, which is $\sim 4.5 \text{ W m}^{-1} \text{ K}^{-1}$ and close to that of the b -axis. Since thermal conductivity scales with the square of sound velocity³⁶, such behavior can be understood by the similar elastic constant and sound velocity along the b - and a -axis, even at high pressures^{33,45}. Therefore, even though we did not measure the pressure dependence of

Λ^{100} , we expect it to be similar to Λ^{010} and much larger than the Λ^{001} . As we show later in the sections of numerical modelling and discussions, what critically impacts slab's thermal evolution is the thermal insulating effect by the low value of Λ^{001} , and the strong thermal conductivity anisotropy. Lack of data for $\Lambda^{100}(P,T)$ does not influence our conclusions of this study.”

9. In the absence of X-ray diffraction to confirm in-situ that the orientation of the single crystal remains stable during compression, and the measurements, is the orientation of the sample in the diamond anvil cell well controlled? how does this contribute to the uncertainty of the thermal conductivity measurement along a specific crystal orientation?

Currently, it is not available to couple the single crystal X-ray (not to mention the electron backscattered diffraction) with time-domain thermoreflectance technique to *in situ* probe the crystal orientation and thermal conductivity, respectively, under high pressure-temperature conditions. We, however, did check the crystal orientation of the quenched antigorite sample out of the DAC after our measurements using electron backscattered diffraction or X-ray diffraction; we found that the orientation remains essentially the same. This indicates that throughout our measurements, we always probed the sample's thermal conductivity along an essentially fixed orientation. Thus the minor variation, if there is, of crystal orientation is expected to have minor effects on the uncertainty of the thermal conductivity.

To clarify the orientation issue, we have added the following sentences in line 146-152:

“By contrast, to check the potential variation of crystal orientation during our compression conditions, we have re-characterized the orientation from the quenched samples out of the DAC using electron backscattered diffraction or X-ray diffraction (*in situ* measurements of crystal orientation and thermal conductivity at high pressures are currently not available). We found that the orientation remained essentially the same, suggesting that the minor variation, if there is, of crystal orientation is expected to have minor effects on the uncertainty of our antigorite's thermal conductivity.”

10. There is no discussion on the possible role of grain boundaries, or the applicability of the single crystal measurements of conductivity to rocks, which are polycrystals. This should be addressed somewhere.

Our single-crystal data along the *b*- and *c*-axis provide important basis that brackets the conductivity of polycrystals (please also see our replies to comment #2 and #4 of reviewer #1). Its applicability to estimate the “average” thermal conductivity of a rock depends on many factors, such as the realistic fraction of crystals along *b*- and *c*-axis,

which, in turn, is influenced by: 1) the grain size; 2) the misorientation of the polycrystalline aggregate with respect to the local stress state; 3) the presence of other phases in the aggregate; 4) the large-scale stress field imposed by the tectonic settings.

It has been reported that the maximum orientation density of the [001] direction along the Z-axis is $\sim 4.7\text{--}18$ m.u.d. (multiples of uniform distribution), and most of the grains orient their [001] within a $\pm 30^\circ$ angle with respect to the Z-axis [see Padrón-Navarta et al. EPSL 2012 (Ref 38) and the references mentioned in our reply to comment #4 of reviewer #1]. As we replied to comment #4 of reviewer #1, the thermal conductivity difference between a fully aligned aggregate and a slightly misaligned one is $\sim 8\text{--}20\%$, whereas its anisotropy remains high ($\sim 2.3\text{--}3.4$). In order to compute the aggregate thermal conductivity of an antigorite mineral assemblage, one has to determine: 1) the distribution density of the [001] orientation with respect to the Z-axis (which should be less than the $\sim 4.7\text{--}18$ m.u.d peak); and 2) the grain size of each crystal. With this information, it is then possible to compute a weighted average considering what percentage of the bulk volume is oriented in which direction. There are codes like D-Rex (Kaminski et al., GJI 158, 744 (2004)) which allow to compute the orientation and the grain size of crystals produced by an external stress field. These codes, however, are limited to olivine and enstatite slip systems. Implementing antigorite slip systems in these codes to produce CPO pole figures is beyond the scope of this paper. Our simplified approach is sufficient to prove the importance of antigorite anisotropy on slab's thermal evolution.

The presence of misaligned grain boundaries can generate additional scattering of the heat-carrying phonons and result in a thermal resistance, i.e., thermal conductivity reduction (Klemens, Intl. J. Thermophysics, 15, 1345 (1994)). The scattering due to tilt boundaries, however, seems to be minor with respect to scattering in the inter-grain regions, which is where the crystallographic disorder concentrates (Klemens 1994). Moreover, it has been shown that the thermal resistance across the basal plane of mica (a phyllosilicate like antigorite) decreases as the planes are forced together by the increasing pressure (Powell and Griffiths, Proc. Roy. Soc. London. Series A-Math. Phys. Sci., 163, 189 (1937)). As suggested by (Wood, Proc. Roy. Soc. London. Series A-Math. Phys. Sci., 163, 199 (1937)), the fall in mica thermal conductivity reported by Powell and Griffiths (1937) is attributed to the grain size reduction due to the crushing of the sample at high pressure. From these experimental evidences and theoretical considerations, it seems that phonon scattering at the grain boundaries of a phyllosilicates aggregate is mostly affected by the grain size rather than the misalignment of the crystals.

For these reasons we argue that the misalignment of antigorite crystals in a serpentinized rock has a minor impact on the aggregate thermal conductivity of the mineral assemblage, because: 1) the misalignment is within a narrow angle ($\pm 30^\circ$); 2) the

misalignment tends to reduce at high pressure and under high shear stress.

It will be interesting to investigate the effects of grain size on antigorite thermal conductivity. Padrón-Navarta et al. EPSL 2012 reports a natural sample of serpentinite with a grain size of few tens of μm , which is at the same order of magnitude of the single crystal samples we measured. If the reduction of thermal conductivity due to smaller grain size is proved experimentally in the future, it would further strengthen our point: a fine-grain aggregate of aligned antigorite crystals produced by an external shear stress field would create an even better thermally insulating layer. For these reasons, we believe that antigorite's anisotropic thermal conductivity plays an important role in influencing the temperature evolution of the slab down to ~ 250 km depth.

To discuss the potential role of grain boundary and polycrystals, we have added a new section in Supplementary Information Text S9: Potential effects of grain boundary.

11. A texture with 100% of crystals oriented with c axis normal to the slab surface is an extreme case. What would be the thermal conductivity of a rock with an intermediate, more realistic, fabric? The effect of CPO is tested here only on two extreme unrealistic cases. This, in my view, does not allow writing the sentence line 363.

This comment is very similar to the comment #4 of reviewer #1 and kind of related to the comment #10 above, so please see our replies there for details and Supplementary Information Text S8.

As we argued earlier, the thermal conductivity of a rock in a slab with a certain realistic fabric depends on the local stress state that the rock experiences and the tectonic settings, as these factors determine the fraction of crystals along the *b*- and *c*-axis, respectively. As we cited in Ref 37 and 38, the almost perfect alignment of single-crystalline antigorite we considered in our model could indeed be present in some subduction systems. Our present results provide novel, crucial data to model the average thermal conductivity of a serpentine rock in a subduction zone, if its local shear stress or serpentine's orientation is known. Most importantly, our findings provide conceptual advance in the impacts of antigorite's *c*-axis thermal insulating effects. If the majority of antigorite aligns along the *c*-axis (which is likely the case during shear deformation), the rock's thermal conductivity will be substantially lower than previously thought with impacts on slab's temperature profile

12. The discussion is only speculative. There is a mix up with the scenario of thermal runaway and melting, lines 370 to 375. Then, lines 376 to 396, the thermal runaway scenario proposed does not explain clearly how antigorite is involved. The ref. 49 is based on a specific olivine rheology, which is different from antigorite rheology, and from its

frictional properties. From figure 8, the thermal runaway seems to occur in antigorite itself (?) (to my knowledge has not been tested numerically). Figure 8 should be clarified accordingly.

To clarify the thermal runaway, we have revised and added the following sentences in line 416-422:

“On the other hand, the seismicity in the lower plane of a DSZ is more enigmatic and under debate. Depending on the variable temperature profile and water content, etc., in a subduction zone, the lower seismic plane could be caused by either also the dehydration of hydrous minerals⁹, or other rupture mechanisms. Another plausible trigger for intermediate-depth earthquakes is the thermal runaway^{12,18-21}, a ductile deformation mechanism in shear zones, where the intense frictional heating induces weakening of rocks and causes the formation of a self-localizing slip plane.”

Through the thermal runaway, the rock within a shear zone could undergo ductile or viscous deformation. If the temperature is heated high enough, melting can occur as evidenced by the glassy pseudotachylites, see Ref^{19,61}.

Moreover, to clarify the role of antigorite in promoting thermal runaway, we have revised the following sentences and added “Ferrand, Lithos 2019” as the Ref 12 in line 429-448:

“Although we have not tested this mechanism in our models, we propose that antigorite’s anisotropic thermal insulating effect can promote the onset of thermal runaway. The ideal setting to trigger thermal runaway are the pre-existing shear zones²¹, in particular the highly serpentinized normal faults produced in trench-rise system. It has been suggested that mineral transformations, such as dehydration of antigorite, in the fault zones would assist the fault slip that generates heat, leading to self-localizing thermal runaway (see Ref¹² and references therein). During shear deformation, the antigorite crystals hosted within the fault zones should orient themselves with the [001] direction perpendicular to the slip plane. In this configuration, the heat produced by friction can flow easily along the slip plane (due to the high λ^{010}), but rarely escapes from the shear zone⁵⁷ due to the thermal blanketing effect by the surrounding antigorite’s low λ^{001} . Thermal runaway is also aided by the weak mechanical resistance of antigorite crystals along the basal planes¹⁶. Moreover, to achieve runaway deformation regime, the background temperature of the shear zone has to be lower than a critical temperature T_c , below which the thermal relaxation generates unstable ductile deformation¹⁸. Typically, the critical temperature of a mantle rock is < 700 K⁶². Our models show that the thermal insulating effect of oriented antigorite keeps the slab interior in colder temperatures, which may be a necessary condition for the onset of thermal runaway. Potentially, the antigorite-assisted thermal runaway could be responsible for the lower seismic plane of a DSZ¹².”

Though the mechanism for lower seismic plane is still under debate, as discussed in Ferrand, *Lithos* 2019 (Ref 12), models of “thermal runaway” and “dehydration-driven stress transfer” can be integrated to explain some intermediate-depth earthquakes, e.g., the seismicity in the lower plane of a DSZ: local dehydration of antigorite in the vicinity of fault zones could trigger and promote the thermal runaway.

Finally, we have also clarified the Fig. 7 (original Fig. 8) by adding and revising the following sentence in the caption regarding the role antigorite plays in promoting thermal runaway in line 393-396:

“Potentially, antigorite can trigger thermal runaway within the pre-existing serpentinized faults by promoting the accumulation of frictional heat within shear fault zones due to its thermal blanketing effect.”

13. In the case such thermal distributions would be real, I wonder whether they should also be seen in seismic velocities or lead to a re-interpretation of some seismic observations?

We can use the scaling parameter reported by Deal et al., *JGR* 104, 28789 (1999) (Ref 59) to approximately convert the temperature difference to compressional velocity V_p perturbation:

$$\frac{dV_p}{dT} \approx 4.8 \times 10^{-4} \left[\frac{km}{s} \frac{1}{^\circ C} \right]$$

This means that, the maximum temperature differences of +170 K and -65 K between model 10 (100% antigorite horizontal insulator) and model 1 (dry olivine) would correspond to a velocity perturbation of $-8.2 \times 10^{-2} [km s^{-1}]$ and $+3.1 \times 10^{-2} [km s^{-1}]$. The V_p in the upper mantle ranges between $\sim 7-10 [km s^{-1}]$ (e.g. Cammarano et al., *PEPI* 138, 197 (2003), Ref 60), and hence the temperature difference found in our models would cause a variation of V_p of $\sim -1\%$ and $\sim +0.4\%$, which should be seismically detectable. Model 10, however, was characterized by a 3-km-thick layer of fully serpentinized rocks, which is an ideal case and not yet found in nature. More realistically, in certain regions of the Earth, the seawater alteration of the oceanic lithosphere can produce a 3-km-thick layer of hydrous rocks containing $\sim 50-70$ vol% of serpentine (e.g. Lesser Antilles, see Allen et al., *JGR* 2022, Ref 55), and Cooper et al. *Nature* 2020, Ref 56). In this study we modelled the case of 50% serpentinization (see model 9, horizontal insulator), which causes a maximum temperature difference of +80 K and -25 K, i.e. $-3.8 \times 10^{-2} [km s^{-1}]$ ($V_p \sim -0.5\%$) and $+1.2 \times 10^{-2} [km s^{-1}]$ ($V_p \sim +0.15\%$). In this case, the seismic anomaly is much harder to observe.

Moreover, the uncertainties in inferring the thermal structure from seismic interpretation should also be considered, which are estimated to be around $\pm 100 K$ in

the upper mantle (Cammarano et al., PEPI 138, 197 (2003)), and comparable to the thermal anomaly produced in our models. Moreover, the perturbation in seismic velocities can be attributed to several factors other than temperature, e.g. 1) intrinsic density of a given mineral phase (antigorite is less dense than olivine); 2) the composite elastic properties of the mineral assemblage; 3) presence of fluid/melts. In our opinion, it will be challenging to attribute univocally a seismic or temperature anomaly to a single factor, such as the presence of an insulating layer inside the slab. Nevertheless, knowing of the possible existence of such layer would help interpret the seismic tomography of pervasively serpentinized subduction zones.

To discuss the relationship between thermal and seismic anomaly, we have added a paragraph in line 399-415:

“Moreover, the temperature differences caused by antigorite’s thermal insulating effect may contribute to local seismic anomalies. Using the scaling parameter reported by Deal et al.⁵⁹, temperature differences of +170 K and –65 K between model 10 and model 1 would lead to a seismically detectable perturbation of compressional velocity V_p of about $-8.2 \times 10^{-2} \text{ km s}^{-1}$ and $+3.1 \times 10^{-2} \text{ km s}^{-1}$, respectively (i.e., about –1% and +0.4% of the V_p in the upper mantle⁶⁰). In the case of 50% serpentinization (relevant to, e.g., Lesser Antilles^{55,56}), temperature differences of +80 K and –25 K would lead to a V_p perturbation of about $-3.8 \times 10^{-2} \text{ km s}^{-1}$ and $+1.2 \times 10^{-2} \text{ km s}^{-1}$ (about –0.5% and +0.15%, respectively, of the V_p in the upper mantle), which may be difficult to clearly observe by seismology. In addition, the uncertainties in inferring the thermal structure from seismic interpretation are estimated to be about $\pm 100 \text{ K}$ in the upper mantle⁶⁰, which is comparable to the thermal anomaly produced in our models. Seismic perturbations, however, can also be attributed to several factors other than temperature, including 1) the intrinsic density of a given mineral phase (antigorite is less dense than olivine), 2) the composite elastic properties of the mineral assemblage, and 3) the presence of fluid/melts, etc. Thus, it would be challenging to attribute univocally a seismic or temperature anomaly to a single factor.”

Less major points:

14. It is interesting, but is not made clear: why testing the influence of thermal conductivity of hydrous olivine (lines 283-290) – if we take the starting point of the manuscript, which is the investigation of the effect of the anisotropic thermal conductivity of antigorite.

Olivine is the major phase of the subducting slab, representing up to 80% of the harzburgitic lithosphere (Irifune and Ringwood, *EPSL* 86, 365 (1987)). At high P - T , olivine is a nominally anhydrous mineral (NAM) that can contain up to few thousands ppm of water (Bolfan-Casanova, *Mineralogical Magazine*, 69, 229 (2005), Ref 48). Therefore, it is likely that olivine could contain part of the water released by the breakdown of hydrous phases. The presence of water inside the slab, however, is confined within 20 km (Gravemeyer et al., *EPSL* 258, 528 (2007)) to 40 km (Faccenda et al., *G3*, 13, Q01010 (2012), Ref 54) from the slab surface. We can therefore call this region, the ‘hydrous layer’ of the slab. As measured by Chang et al., *PNAS* 2017 (Ref 40), water reduces the thermal conductivity of olivine by ~30-40% at the transition zone depth, and hence olivine can potentially also create a thermal insulating layer inside the slab. By comparing the antigorite-bearing models with a heavily hydrated olivine case (~7000 ppm wt water), we wanted to estimate the potential of antigorite’s thermal insulating properties. In our models, we showed that a 3-km-thick layer of 50% serpentinized rock (50% antigorite + 50% dry olivine) with the [001] direction oriented perpendicularly to the slab surface (model 9), has an insulating effect that is comparable to a 3-km-thick layer of water saturated olivine (Figure 5 in the main text). However, at the P - T conditions of the trench and upper mantle, olivine has a water solubility of only few hundreds wt. ppm (Bolfan-Casanova, *Mineralogical Magazine*, 69, 229 (2005), Ref 48) with relatively high thermal conductivity. Thus antigorite with [001] direction would be a key mineral phase to act as a thermal blanket in the shallow upper mantle (< 230 km). With this comparison, we want to stress the importance of antigorite as a thermal insulator in the shallow upper mantle.

To comment on this issue, we have added the following sentence in line 306-309:

“Given the limited water storage capacity of olivine in the upper mantle (only about few hundreds wt. ppm⁴⁸) with relatively high thermal conductivity, antigorite with [001] direction would be a key mineral phase to act as a thermal blanket in the shallow upper mantle.”

15. In the methods section line 453-454, the reader is only referred to previous publications for P and T uncertainties, and there are no error bars, or information on the uncertainty, on figure 1 and 2 for the pressure (and temperature fig. 2). For general readers, this could be the first paper they read, using this experimental technique. Please provide some information in the main text or in the methods section, on the uncertainties for P .

In our high pressure and room temperature measurements, the highest pressure was ~13 GPa, where the pressure uncertainty was only ~0.1-0.2 GPa. In the high pressure-

temperature measurements, the pressure uncertainty was ~0.1-0.5 GPa due to the broadening of ruby peak. To clarify this, we have added the following information in

(1) line 504:

“with an uncertainty of ~0.1–0.2 GPa”

(2) in line 511-512:

“The pressure uncertainty is ~0.1–0.5 GPa due to the broadening of ruby peak upon heating.”

16. Figure 7 could have arrows hotter/colder for instance, to facilitate understanding as first sight.

We agree with the reviewer, and have added the hotter/colder arrows to the original Figure 7, which is now changed to Figure 6.

Line by line comments:

17. Line 42: ref. 4, I believe there are anterior references for this.

There are. But we wanted to cite a recent paper where they modelled the thermal evolution of the slab, and where they compared the dehydration depth of antigorite with the distribution of intermediate-depth seismicity. Wei et al., 2017 approach is in line with what was done in our work, and it also supports our argument, i.e., ‘the temperature profile of the subduction zone is key to influence the stability of the minerals and rocks within it, as well as its geological processes, e.g., arc volcanism and magmatism and earthquakes.

18. Introduction lines 46 from 53: the publications listed here are mostly from non-experimental work, while the literature is abundant on this topic. Experimental works actually played a very important role, if not the most important, in proposing all these mechanisms.

To provide more experimental works (but limited by the total reference number), we have added

- (1) Jung, H., Green II, H. W., & Dobrzhinetskaya, L. F. (2004). Intermediate-depth earthquake faulting by dehydration embrittlement with negative volume change. *Nature*, 428(6982), 545-549, as Ref 11 for dehydration embrittlement.
- (2) Ogawa, M. (1987). Shear instability in a viscoelastic material as the cause of deep focus earthquakes. *Journal of Geophysical Research: Solid Earth*, 92(B13), 13801-13810, as Ref 20 for thermal runaway.

19. Line 51: Reynard, B., J. Nakajima, and H. Kawakatsu (2010), *Earthquakes and plastic deformation of anhydrous slab mantle in double Wadati-Benioff zones*, *Geophys. Res. Lett.*, 37, L24309, doi:10.1029/2010GL045494 may be cited also here.

We have added the paper by Reynard et al. GRL 2010 the reviewer refers to as the Ref 16.

20. Line 56, ref 20 unnecessary, while many other would be relevant.

Following the reviewer's suggestion, we have removed the original Ref 20 "Liang et al., 2020".

21. Line 63: ref. 7 could also be cited here.

We have added Yamasaki & Seno, JGR 2003: Double seismic zone and dehydration embrittlement of the subducting slab, now Ref 6 to line 64.

22. Line 128: ref. 62 could also be cited here.

We have added Ref 62 (now Ref 43-Hilairret et al., GRL 2006: Equation of state of antigorite, stability field of serpentines, and seismicity in subduction zones) to line 136.

23. Line 182-183: there should be a reference to the supplementary material for more information.

We have added a new sentence in line 217-220 as reference explicitly to the supplementary material:

"The full description of the model is reported in Supplementary Information: physical and numerical model (Text S1-S2, Fig. S4-S8, Table S1), thermodynamic parameters (Text S3-S5, Fig. S9, Tables S2-S3), model limitations (Text S6), and code benchmark (Text S7, Fig. S10-S11, Table S7)."

24. Lines 213-215: I do not understand the connection between the first and second part of this sentence (before and after "hereafter").

To avoid confusion and make the numerical model results section more readable, we have removed these sentences.

25. Lines 286: I am not sure what 'similarly' refers to (did you mean the differences between wet and dry are similar to the differences between dry olivine and models that include antigorite?)

This sentence was removed to make the numerical results section more readable.

26. Line 304: *“moreover” and “further” is kind of repeating.*

We have removed the “moreover” from line 310 (former line 304).

27. Line 304-305: *I find “antigorite stability field” clumsy here, because for mineral physicists, stability field refers to a P-T stability field in the thermodynamic sense, a property of the mineral which will not vary - “breakdown depths”, as used later in the text, is more appropriate to me.*

We have replaced the “stability field” with “breakdown depth” in line 311 (former line 305).

28. Lines 349 to 353: *this scenario is presented like a certainty with only one reference, while I believe it is debated.*

In line 353-357, we have re-formulated our statement, where we now address it as a hypothesis:

“Potentially, the unbending of the slab in the mantle could generate a pressure gradient sufficient to pump seawater toward the inner portion of the slab, and to push the serpentinization front down to ~40 km from the slab surface⁵⁴. The occurrence of this serpentinization process, however, is still under debate.”

29. Lines 366-368 *there is only one reference here - it would be good to have additional references, including modeling and mineral physics ones.*

In our Discussion section, we have toned down and revised our discussions about the mechanisms to trigger seismicity in the upper and lower plane of a DSZ to the following sentences in line 416-428:

“On the other hand, the seismicity in the lower plane of a DSZ is more enigmatic and under debate. Depending on the variable temperature profile and water content, etc., in a subduction zone, the lower seismic plane could be caused by either also the dehydration of hydrous minerals⁹, or other rupture mechanisms. Another plausible trigger for intermediate-depth earthquakes is the thermal runaway^{12,18-21}, a ductile deformation mechanism in shear zones, where the intense frictional heating induces weakening of rocks and causes the formation of a self-localizing slip planes. To produce such slip planes, the rate of heat production inside the shear zone has to be larger than the rate of heat dissipation from the shear zone¹⁸. When this condition occurs, the net heat confined inside the shear zone leads to a temperature increase, which promotes ductile deformation and

generates weak slip planes. The severe frictional heat sometimes can lead to the formation of lenses of molten rock (e.g., pseudotachylites^{19,61}).”

Here we have added a paper by “Ferrand, Lithos 2019” as the Ref 12, and also removed the original sentence “The lower plane, however, occurs around the coldest portion of the slab, well within the stability field of most hydrous minerals” that the reviewer commented.

30. Conclusion, line 425: I believe the influence of DHMS will depend on their amount at depths.

The reviewer is correct that the effects of DHMS on the thermal evolution of slab will depend on their amount at depths, which remains uncertain. To avoid confusion, we have removed the DHMS.

To end our manuscript better, we have also revised the sentences in line 474-482 to: “Future experimental studies on the thermal conductivity of relevant phyllosilicates in the sediments and crust (e.g., muscovite, biotite, chlorite, phengite, and talc, etc.) along specific crystal orientation and *P-T* conditions of slab subduction will offer more comprehensive datasets for the thermal properties of slab’s minerals. Combined with detailed thermal evolution modelling, these would substantially advance our understanding of important geological, geophysical, and geodynamical features in a subduction zone, including the thermal state, flow dynamics, water transportation, magmatism, and seismic structure of the region.”

31. Figure S5: the single-ended arrows: what do they mean?

We have added the following sentence to the caption of Figure S6 (former Figure S5):

“The red arrows indicate the heat flow from the hot ambient mantle toward the cold slab.”

REVIEWER COMMENTS

Reviewer #4 (Remarks to the Author):

Authors measured experimentally that antigorite's thermal conductivity is strongly anisotropic, and they showed numerically that this anisotropy can affect the thermal structure of the slab with potential implications for the breakdown depth of the hydrous minerals. I think that their experimental and numerical findings present significant conceptual advances in revealing the impacts of antigorite on the thermal evolution of slab and the occurrence of intermediate-depth earthquakes. The results are interesting. The paper is well written and organized. My comments for improvement of the paper are as follows.

1) Lines 362-364. Regarding on the description of crystal preferred orientation (CPO) of antigorite, authors need to correct the statement "the [001] direction tends to align perpendicularly to the shear direction". This statement is partly true. Jung (2011) showed that the [001] axis of antigorite from Val Malenco and Punta Bettolina in NW Italy are aligned subnormal to the foliation of serpentinite. There are more evidences of this kind of CPOs of antigorite reported in the recent review paper (Jung, 2017). In the line 364, therefore, it is suggested to change from "shear direction" to "shear plane / foliation".

2) Line 51. Authors are suggested to add a recent reference here about the dehydration embrittlement of chlorite for intermediate-depth earthquakes (Kim et al., 2023).

3) Lines 417-420. Regarding on the discussion of the seismicity in the lower plane of a double seismic zone(DSZ), there is a recent paper which demonstrated experimentally the occurrence of faults by dehydration embrittlement of chlorite at the high pressure and temperature conditions corresponding to the lower plane of DSZ in slabs (Kim et al., 2023).

References

Jung, H. (2011). Seismic anisotropy produced by serpentine in mantle wedge, *Earth and Planetary Science Letters*, 307, 535-543.

Jung, H. (2017). Crystal preferred orientations of olivine, orthopyroxene, serpentine, chlorite, and amphibole, and implications for seismic anisotropy in subduction zones: a review, *Geosciences Journal*, 21, 985-1011.

Kim, D., Jung, H., and Lee, J. (2023). Impact of chlorite dehydration on intermediate-depth earthquakes in subducting slabs, *Communications Earth & Environment*, 4, 491.

Haemyeong Jung

Reviewer #5 (Remarks to the Author):

The manuscript by Chien and co-authors present new experimental data on antigorite thermal conductivity.

Having no background in experimental petrology, I won't comment on the soundness/quality of the experimental methodology and results. I will assume this part is correct.

I would expect the first figure of the paper to show an example of a DSZ, but also (maybe as a later figure) how thermal modeling simulations are able to fit the observables.

Overall, I find that the first order modelling results indicate a realistic thermal blanketing effect due to anisotropic thermal conductivity. Even if the thermal modelling study could have been performed more easily using existing thermo-mechanical code (including forward modeling of subduction), the 2D finite difference thermal code presented in this study, is exhaustively described and appropriate. However, within the framework of open-source science the matlab code should be provided to allow data reproducibility.

I also think that a compilation of subduction zones where DSZ are observed should be presented. If conductivity anisotropy of antigorite may be a possible explanation for the DSZs, I would expect to find correlation between subduction where DSZ is observed some properties of the slab. For instance, is the distance between the two seismogenic layers of the DSZ depends on the subduction system? Is it possible to correlate this distance based on the inferred hydration depth of the slab?

I think the manuscript does present a very interesting explanation for the DSZ. However, in my opinion it would be important to try to relate the modeling findings of the study with observable of the DSZ.

Nicolas Riel,

L350-351: The statement that hydration at ridge produces a few km of serpentine is at best misleading. To my knowledge this is clearly not the case in all systems and may only be true for slow spreading centers where the mantle is exposed. However, the author should consider deep hydration processes during bending of the plate and inverted fluid pressure gradient (Faccenda 2009, Deep slab hydration induced by bending-related variations in tectonic pressure)

In what follows, both reviewers' comments are in italics and our point-to-point responses are in normal fonts with blue color. Changes in our revised manuscript are also labeled by blue color. We have added additional information and made revisions accordingly, which have fully addressed the reviewers' suggestions and questions.

Report of Referee # 4

Authors measured experimentally that antigorite's thermal conductivity is strongly anisotropic, and they showed numerically that this anisotropy can affect the thermal structure of the slab with potential implications for the breakdown depth of the hydrous minerals. I think that their experimental and numerical findings present significant conceptual advances in revealing the impacts of antigorite on the thermal evolution of slab and the occurrence of intermediate-depth earthquakes. The results are interesting. The paper is well written and organized. My comments for improvement of the paper are as follows.

We thank the reviewer for applauding and appreciating our important findings in this paper. We have addressed your suggestions as follows to improve our manuscript.

1) Lines 362-364. Regarding on the description of crystal preferred orientation (CPO) of antigorite, authors need to correct the statement "the [001] direction tends to align perpendicularly to the shear direction". This statement is partly true. Jung (2011) showed that the [001] axis of antigorite from Val Malenco and Punta Bettolina in NW Italy are aligned subnormal to the foliation of serpentinite. There are more evidences of this kind of CPOs of antigorite reported in the recent review paper (Jung, 2017). In the line 364, therefore, it is suggested to change from "shear direction" to "shear plane / foliation".

Following the reviewer's suggestion, we have changed the "shear direction" to "shear plane/foliation" in line 368. We have also added "Jung, H. (2017), Crystal preferred orientations of olivine, orthopyroxene, serpentine, chlorite, and amphibole, and implications for seismic anisotropy in subduction zones: a review, *Geosciences Journal*, 21, 985-1011" as the new Ref 59 in line 368.

2) Line 51. Authors are suggested to add a recent reference here about the dehydration embrittlement of chlorite for intermediate-depth earthquakes (Kim et al., 2023).

We thank the reviewer for bringing out this important paper. We have added a recent paper by Kim *et al.*, Impact of chlorite dehydration on intermediate-depth earthquakes in

subducting slabs, *Commun Earth Environ* **4**, 491 (2023) as the new Ref 13 in line 55.

3) Lines 417-420. Regarding on the discussion of the seismicity in the lower plane of a double seismic zone (DSZ), there is a recent paper which demonstrated experimentally the occurrence of faults by dehydration embrittlement of chlorite at the high pressure and temperature conditions corresponding to the lower plane of DSZ in slabs (Kim *et al.*, 2023).

Following the reviewer's suggestion, we have added a recent paper by Kim *et al.*, Impact of chlorite dehydration on intermediate-depth earthquakes in subducting slabs, *Commun Earth Environ* **4**, 491 (2023), the new Ref 13, in line 426.

Report of Referee # 5

1. The manuscript by Chien and co-authors present new experimental data on antigorite thermal conductivity. Having no background in experimental petrology, I won't comment on the soundness/quality of the experimental methodology and results. I will assume this part is correct. I would expect the first figure of the paper to show an example of a DSZ, but also (maybe as a later figure) how thermal modeling simulations are able to fit the observables.

We thank the reviewer for raising this suggestion. To illustrate basic ideas and some examples of a DSZ, in the very early stage of the Introduction in line 51, we have already provided some references (e.g., Ref 4-6) to the readers. To guide the readers to better understand example features of DSZs, we have added “, see Ref ⁴⁻⁶ for example features of DSZs, where the separation of two seismic planes correlates with the age of a slab, i.e., with its temperature and thickness at the trench.” in line 51-53. (Also see our replies to the comment #3 below.)

We actually showed an example illustration of a DSZ and the correlation between its seismic planes and our thermal model simulations in the Fig. 7. We think that making this figure in the discussion section is more coherent with the logic and flow when we present our results and discussions. We thus prefer to keep our current form and order of the figures (rather than showing an example DSZ in the first figure).

2. Overall, I find that the first order modelling results indicate a realistic thermal blanketing effect due to anisotropic thermal conductivity. Even if the thermal modelling study could have been performed more easily using existing thermo-mechanical code (including forward modeling of subduction), the 2D finite difference thermal code

presented in this study, is exhaustively described and appropriate. However, within the framework of open-source science the matlab code should be provided to allow data reproducibility.

We agree with the reviewer about the open source for scientific research. In line 605-607, we have added a “Code Availability” section with “The MATLAB script used to perform numerical modelling is available in <https://zenodo.org/records/11104058>.” to explicitly let colleagues know where to download this script. The code is open source under MIT license (<https://opensource.org/license/mit>). By downloading the main code and its functions it will be possible to reproduce our models from every computer with MATLAB 2022 installed.

We believe that our simple thermal diffusion code is actually more suited for investigating the possible large-scale implications arising from our thermal conductivity measurements. The advantages of this code rely on its simplified model setting and physical description. By computing exclusively P - T -dependent heat diffusion, the model is only susceptible to thermal conductivity changes, which is the focus of this paper.

Combining mineral physics data with numerical modelling is quite challenging, because the realistic modelling of a rock should include: 1) the most abundant minerals, 2) their physical properties, and 3) the dependency of these properties to thermodynamic variables (temperature, pressure, chemical composition, etc.). Given the large number of thermodynamic parameters and their dependencies, it is quite difficult to disentangle the contribution of each mineral from a complex, although more realistic, thermo-mechanical model.

Moreover, in a thermo-mechanical model there are several non-linear phenomena that heavily affect the outcome of the simulation. For example, thermal conductivity causes changes in temperature, which in turn causes changes in density and viscosity, thus altering the subduction velocity of the slab and its rheology. These further affect the temperature evolution of the rocks and their thermal conductivity. In our work we limited this physical description down to one non-linearity: thermal conductivity causes changes in temperature, which causes changes in thermal conductivity during subduction. In this way we can have a better focus on the measured parameter, and its P - T -crystal orientation dependencies.

Many papers draw their conclusions and implications directly from other mineral physics measurements. In our workflow, instead, we preferred to include simple, yet reliable models to upscale our laboratory measurements to large-scale geological settings. In this way, within the same paper, we proved the importance of our experimental measurements to the numerical modelling community. In our conclusions, in fact, we

encouraged colleagues to include our findings in future more complex thermo-mechanical models of slab subduction.

3. *I also think that a compilation of subduction zones where DSZ are observed should be presented. If conductivity anisotropy of antigorite may be a possible explanation for the DSZs, I would expect to find correlation between subduction where DSZ is observed some properties of the slab. For instance, is the distance between the two seismogenic layers of the DSZ depends on the subduction system? Is it possible to correlate this distance based on the inferred hydration depth of the slab?*

I think the manuscript does present a very interesting explanation for the DSZ. However, in my opinion it would be important to try to relate the modeling findings of the study with observable of the DSZ.

We thank the reviewer for pointing out the possible correlation between the separation of seismic planes in DSZs and the dehydration depth of the slab. We first clarify that as we emphasized in the manuscript, our findings suggest antigorite's anisotropic thermal conductivity combined with CPO promotes dehydration embrittlement and thermal runaway (both mechanisms could play key roles in triggering intermediate-depth earthquakes). However, the exact mechanism causing the lower seismic plane of a DSZ remains debated, since many factors (e.g., age, temperature, dipping angle, sinking velocity, hydration content, and tectonic setting, etc.) of a subduction zone would affect the condition to trigger an intermediate-depth earthquake. Thus, we did not unequivocally conclude that it is the antigorite's dehydration causing both upper and lower seismic planes, since they could be caused by other effects. Nevertheless, our findings here clearly demonstrated the importance of antigorite's CPO and anisotropic thermal conductivity that significantly promotes the potential mechanisms triggering intermediate-depth earthquakes and even DSZs.

A compilation of 16 different subduction zones featuring DSZ has actually been presented in the Ref 4 [Brudzinski *et al.*, *Science* **316**, 1472–1474 (2007)] when we introduced the DSZ in line 51, which should have provided the readers with sufficient background about the global prevalence and features of DSZs.

The reviewer is correct: there is a correlation between the separation of the two seismic planes of a DSZ, S_{DSZ} [km], and the physical properties of the subduction zones. In particular, the S_{DSZ} depends approximately linearly on the age of the subducting plate, see Ref 4, Brudzinski *et al.*, *Science* 2007. We report the S_{DSZ} as a function of the slab's age from Brudzinski *et al.* *Science* 2007 in the figure below:

We here also computed the thermal parameter for the 30 slab segments reported by Brudzinski *et al.* Science 2007: $\phi = \text{age} * \text{convergence_velocity} * \sin(\text{dip_angle})$. In the figure below, we plot the S_{DSZ} as a function of the thermal parameter:

In this case, the data are less correlated. Therefore, it seems that the separation of planes depends mostly on the slab age, i.e., the temperature state and the thickness of the slab at the trench.

To provide more information about the features of DSZs, in line 51-53, we have added “, see Ref ⁴⁻⁶ for example features of DSZs, where the separation of two

seismic planes correlates with the age of a slab, i.e., with its temperature and thickness at the trench.”

The temperature profile of a slab is characterized by a cold core ($T < 700$ K), surrounded by hot outer margins ($T > 1000$ K), see, e.g., Fig. 4 in our manuscript. The cold core is ~50-km-thick in old slabs [e.g., Tohoku slab, 140 Myrs, see Fig. 5 in Hacker *et al.*, JGR 108, 2030 (2003)], and ~30-km-thick in young slabs (e.g., Costa Rica slab, 16 Myrs, see Fig. 4 in the same paper by Hacker *et al.*, JGR 2003). Such ~20 km difference can explain the S_{DSZ} distribution as a function of age. It seems that the two planes of seismicity occur in the thermal regions at the margin of the cold core of the slab (700-1000 K). Therefore, the S_{DSZ} should be related to the thickness of the cold core of the slab.

According to many studies, including Hacker *et al.*, JGR 2003, the upper plane of seismicity is almost always located within ~10 km from the slab surface, and is usually attributed to the dehydration of antigorite (Brudzinski *et al.*, Science 2007). As we mentioned earlier and in the manuscript, the exact mechanism responsible for the lower plane of seismicity, instead, is still under debate. Two potential candidates are: 1) chlorite dehydration, occurring at ~900-1100 K (see, e.g., Kim *et al.*, 2023, the newly added Ref 13), and 2) thermal runaway occurring at < 700 K. Our numerical models (see Fig. 6 in the main manuscript) demonstrate that the presence of a thermal insulating layer of antigorite has two effects: 1) facilitating the dehydration of the outer 10 km of the slab (0-10 km), and 2) maintaining the slab core at a relatively cold condition. These two effects combined should enhance the separation between the two planes of seismicity.

To relate our modeling findings with observables of DSZs, we have added “Such effect is expected to enhance the separation of the two seismic planes of DSZs, and should be taken into account to better understand the features of global DSZs⁴.” in line 388-390.

On the other hand, we wish we could do more, but so far we are not able to fully compare our models with the estimates of S_{DSZ} and serpentinization degree μ [%]. For example, in the manuscript we reported the serpentinization degree for the Lesser Antilles trench (90 Myrs old), which has a $\mu \sim 50$ -70% (Refs 57 and 58: Allen *et al.*, 2022 and Cooper *et al.*, 2020). Laigle *et al.*, Tectonophysics, 603, 1-20 (2013) reported a DSZ in the Lesser Antilles trench with a separation of ~18 km. This value is smaller than the 30 km separation in the Tohoku slab (140 Myrs old, see Brudzinski *et al.* Science 2007), for which, however, μ estimates are currently *not available*. Therefore, the comparison between the Lesser Antilles and the Tohoku slab can only confirm the relation between the slab age and S_{DSZ} , but not between μ and S_{DSZ} .

We finally note that the serpentinization degree μ of subducting plates is usually estimated from: 1) the 2D P-wave models of the subduction zone (e.g. Ref 57, Allen *et*

al., 2022); and 2) the magnetic properties of the rocks (e.g. Maffione *et al.*, 2014). The hydration of (Mg,Fe)SiO₄ olivine, in fact, produces Mg-rich serpentine (characterized by low density and low bulk modulus), and magnetite. These estimates are limited for two reasons: 1) most literature investigate the serpentinization of the mantle wedge on top of the dehydrating slab (no information about the original serpentinization degree of the slab), and 2) the estimates are usually scattered depending on which section of the slab is considered or even on which inversion model is used to interpret the seismic data.

L350-351: The statement that hydration at ridge produces a few km of serpentine is at best misleading. To my knowledge this is clearly not the case in all systems and may only be true for slow spreading centers where the mantle is exposed. However, the author should consider deep hydration processes during bending of the file and inverted fluid pressure gradient (Faccenda 2009, Deep slab hydration induced by bending-related variations in tectonic pressure).

We partially agree with the reviewer: serpentinization degree is inversely proportional to the spreading rate of the ridge (Faccenda Tectonophysics 2014, Water in the slab: A trilogy). Nevertheless, the higher degree of serpentinization of slow-spreading ridges is fostered by the presence of long-lived detachment faults that expose large volumes of harzburgitic mantle, and is aided by hydrothermal cooling (Minshull *et al.*, Geological Society, London, 1998, Is the oceanic Moho a serpentinization front?). Serpentinization, in fact, occurs below ~600 °C and it can continue at temperatures below 100 °C (weathering), thus requiring a cold oceanic lithosphere. This thermal status is readily achieved when hydrothermal circulation is active. Moreover, a fast-spreading ridge (2 cm/yr) can also present non-negligible serpentinization (>10%) in the upper 7 km of the oceanic lithosphere, i.e., in the oceanic crust (Iyer *et al.*, EPSL 2010, Feedbacks between mantle hydration and hydrothermal convection at ocean spreading centers). (The definition of lithosphere includes both the crust and the mantle.)

For these reasons we believe that our original statement in line 353 “*The hydrothermal circulation is active at the spreading ridges, and produces a few-km thick layer of serpentinized rocks inside the oceanic lithosphere*” should be reasonable. Nevertheless, to avoid confusion, we have rephrased our statement in line 353-355 to:

“The hydrothermal circulation active at the slow-spreading ridges can reach the bottom of the oceanic crust, and thus produce a few-km thick layer of serpentinized rocks within the lithospheric mantle^{50,51}.”

Following the reviewer’s suggestion, to consider the deep hydration induced by bending of slab, we have added “Faccenda *et al.*, Nature Geoscience 2, 790 (2009), Deep

slab hydration induced by bending-related variations in tectonic pressure” as the new Ref 55 in line 358.

REVIEWERS' COMMENTS

Reviewer #4 (Remarks to the Author):

I think that revision of the paper was well done.

Authors measured experimentally that antigorite's thermal conductivity is strongly anisotropic, and they showed numerically that this anisotropy can affect the thermal structure of the slab with potential implications for the breakdown depth of the hydrous minerals. I think that their experimental and numerical findings present significant conceptual advances in revealing the impacts of antigorite on the thermal evolution of slab and the occurrence of intermediate-depth earthquakes. The paper is well written and organized.

Haemyeong Jung

Reviewer #5 (Remarks to the Author):

The authors did a very careful and detailed revision of the manuscript and answered all the comments I had.

I have no further comments.